# Epigenetic regulation of cocaine intake through dopaminergic control of cholinergic interneurons in male mice

Robert G. Lewis [1,2,7], Lauren Otsuka[1,2,7], Daniela Punzo[1,2], Yasmine Sherafat[3,6], Ermanno Florio[1,2], Mingqi Zhou [4], Valeria Lallai[3], Thu Dinh Nha Pham[1,2], Marcus Seldin [4], Christie D. Fowler [3] & Emiliana Borrelli [1,2,5] ✉

Substance use disorders are chronic neuropsychiatric conditions influenced by multiple factors, shaping individuals' vulnerability to addictive drugs like cocaine. Here, we reveal that dopamine D2 receptor mediated inhibition of striatal cholinergic interneurons regulates the motivation for cocaine intake by modulating acetylcholine signaling in striatal circuits. This acetylcholine-dependent mechanism contributes to cocaine self-administration through the muscarinic M4 receptor and the histone acetyltransferase Kat5/Tip60. We discover that Kat5 is upregulated in response to cocaine and further show that this leads to acetylation of histone H4 on lysine 8, an epigenetic modification that increases immediate early gene expression and muscarinic M4 receptor in dopamine D1 receptor-expressing medium spiny neurons. Notably, this chain of events is absent in male mice lacking D2 receptor-mediated inhibition of cholinergic interneurons, resulting in reduced cocaine consumption. These findings expand our understanding of how cocaine manipulates striatal circuits to reinforce drug-seeking behavior.

Substance use disorders (SUDs) affect millions of people in the US, generating enormous costs associated with the loss of productivity, crime, and healthcare burden[1,2]. Psychostimulants, including cocaine, represent a significant portion of drug use and contribute to almost 15% of overdose-related deaths. Cocaine use induces long-term changes in gene expression in the nucleus accumbens (NAcc)[3,4], which alter the cellular environment and enhance synaptic strength[5]. The underlying mechanisms of these changes include post-translational modifications of histones in a manner independent from an individual's genetics[6,7]. Indeed, epigenetic modifications of chromatin structure in the NAcc are thought to contribute to the development and maintenance of SUDs[8,9].

The NAcc is comprised of 90–95% medium spiny neurons (MSNs) and 5–10% interneurons[10]. MSNs serve as the sole output neurons of the striatum, with projections to the substantia nigra, including also NAcc-specific projections to the ventral tegmental area and the ventral pallidum[11,12]. MSNs are mostly divided into those expressing the dopamine D1 receptor (D1R; D1R+ MSNs) and those expressing the dopamine D2 receptor (D2R; D2R+ MSNs), although a minimal percentage expressing both receptors has also been reported[13]. The increase in DA levels following cocaine intake both stimulates D1R+ MSNs while inhibits D2R+ MSNs activities. Simultaneously, DA exerts a dual influence on the activity of striatal cholinergic interneurons (ChIs), which express both the stimulatory dopamine D5 receptor (D5R) and the inhibitory D2R. DA neurons can also release glutamate, and this co-release has also been shown to regionally modulate ACh signaling in the striatum[14]. Subsequently, acetylcholine (ACh) provides

[1]INSERM U1233, Center for Epigenetics and Metabolism, University of California, Irvine, CA, USA. [2]Department of Microbiology and Molecular Genetics, University of California, Irvine, CA, USA. [3]Department of Neurobiology and Behavior, University of California, Irvine, CA, USA. [4]Department of Biological Chemistry, University of California, Irvine, CA, USA. [5]Department of Pharmacological Sciences, University of California, Irvine, CA, USA. [6]Present address: Department of Psychology, California State University, San Marcos, CA, USA. [7]These authors contributed equally: Robert G. Lewis, Lauren Otsuka. ✉e-mail: borrelli@uci.edu

feedback on MSN activity through the activation of muscarinic receptors on these neurons[15–19]. This establishes a circuitry linking DA and ACh signaling in the striatum upon cocaine use.

Recent studies suggest that the regulation of ChIs plays a critical role in driving the motivating effects of cocaine, potentially contributing to susceptibility to cocaine addiction[18–20]. Mice lacking D2R in ChIs (ChI-D2RKO mice) show attenuated motor responses to acute cocaine, reduced conditioned place preference (CPP), and lack of induction of Fos expression in D1R$^+$ MSNs in contrast to wild-type (WT) controls[18]. Previously, we showed that muscarinic M4 receptor (M4R) antagonism effectively reverses this phenotype[18], indicating that the communication between striatal DA and ACh signaling to MSNs is accountable for these effects. However, the precise molecular mechanisms underlying changes in cocaine-induced behaviors and gene expression in the absence of D2R signaling in ChIs remained to be fully elucidated.

In this study, we show that D2R-mediated control of ChIs is vital for maintaining cocaine-induced motor responses and motivation for cocaine intake. Using male ChI-D2RKO mice, we found unaltered reward learning, but decreased motivation for intravenous cocaine infusions under a progressive ratio (PR) schedule of reinforcement. Transcriptomic analysis of the NAcc of these mice implicated the histone acetyltransferase, Kat5/Tip60[21,22] (hereafter referred to as Kat5), in cocaine-induced gene expression changes. Notably, histone H4 acetylation in the mesolimbic system has been associated with SUDs[23–25], and cocaine-induced expression of immediate early genes (IEGs), such as Fos and Fosb, has been linked to the hyperacetylation of histone H4 on lysine 8 (H4K8ac) at the promoters of these genes[24,26,27]; a hallmark feature of the drug's rapid effects. However, the signaling mechanisms governing acetylation at this residue and the specific acetyltransferase(s) involved have yet to be determined.

Here, we show that in male WT mice, cocaine increases Kat5 induction and H4K8ac levels in the NAcc, indicating an important link between these two molecules. Interestingly, this cocaine-induced stimulation of Kat5 and H4K8ac is absent in ChI-D2RKO mice, together with downregulation of IEG expression as compared to WT littermates. Importantly, we found that the expression of a gene critical for D1R$^+$ MSN activity, M4R, is also regulated by cocaine through this same mechanism exclusively in the NAcc of WT mice. Finally, positive modulation of M4R activity attenuates cocaine intake in WT mice, mirroring the behavior of ChI-D2RKO mice and providing valuable insights into potential therapeutic strategies for cocaine use disorder.

## Results

### Absence of D2R signaling in ChIs leads to reduced motivation to self-administer cocaine

Our previous studies showed that male ChI-D2RKO mice exhibit reduced motor and cellular responses to cocaine administration compared to controls, despite similar baseline motor ability[18]. In addition, ChI-D2RKO mice do not sensitize to cocaine nor do they show the incentive motivational properties of the drug as analyzed by CPP, strongly indicating that altered D2R-mediated signaling in ChIs largely impacts cocaine's effects[18].

To further elucidate the long-term consequences of this phenotype, we analyzed the response of male ChI-D2RKO mice to intravenous cocaine self-administration (IVSA). To first establish that ChI-D2RKO and WT control (Drd2 floxed male littermates) mice can learn an operant task and respond for reward, mice were examined for self-administration of food pellets. Both genotypes similarly learned the task by progressively acquiring a significant number of food pellets across daily sessions (two-way repeated measures ANOVA, session: $F_{(2.993,41.90)} = 22.15$, $P < 0.0001$; genotype: $F_{(1,14)} = 0.4823$, $P = 0.4987$; session × genotype: $F_{(7,98)} = 1.578$, $P = 0.1510$) (Supplementary Fig. 1a and Supplementary Dataset S1a).

Furthermore, subjects were able to differentiate their responses between the levers and demonstrated a statistically significant preference for the active lever as compared to the inactive one (three-way repeated measures ANOVA, session: $F_{(3.164,88.59)} = 35.19$, $P < 0.0001$; genotype: $F_{(1,28)} = 0.5062$, $P = 0.4827$; lever: $F_{(1,28)} = 108.0$, $P < 0.0001$; session × genotype: $F_{(7,196)} = 0.7318$, $P = 0.6452$; session × lever: $F_{(7,196)} = 28.54$, $P < 0.0001$; genotype × lever: $F_{(1,28)} = 0.3172$, $P = 0.5778$; session × genotype × lever: $F_{(7,196)} = 0.2900$, $P = 0.9573$) (Supplementary Fig. 1b and Supplementary Dataset S1b). These results provide evidence of a similar ability of both genotypes to not only perform the task but also in lever pressing behavior, as no genotype differences were found in responding to the active or inactive lever. Male WT and ChI-D2RKO mice were then implanted with intravenous catheters after food training, and an acquisition dose of cocaine (0.3 mg/kg per infusion) was introduced at the established FR5TO20 schedule of reinforcement[28–30]. Both genotypes similarly acquired the task and demonstrated stable responses across daily sessions (Supplementary Fig. 1c and Supplementary Dataset S1c). While a statistical main effect was found for the session, this did not result in any statistically significant difference between groups in the post-hoc test (two-way repeated measures ANOVA, session: $F_{(2.543,30.52)} = 3.165$, $P = 0.0455$; genotype: $F_{(1,12)} = 1.232$, $P = 0.2888$; session × genotype: $F_{(4,48)} = 1.275$, $P = 0.2929$). When the active versus the inactive lever pressing during cocaine acquisition was further assessed, both genotypes exhibited a similar preference for the active lever across sessions (three-way repeated measures ANOVA, session: $F_{(4,96)} = 2.610$, $P = 0.0402$; genotype: $F_{(1,24)} = 0.8339$, $P = 0.3702$; lever: $F_{(1,24)} = 50.45$, $P < 0.0001$; session × genotype: $F_{(4,96)} = 1.447$, $P = 0.2245$; session × lever: $F_{(4,96)} = 3.434$, $P = 0.0114$; genotype × lever: $F_{(1,24)} = 2.089$, $P = 0.1613$; session × genotype × lever: $F_{(4,96)} = 1.389$, $P = 0.2434$) (Supplementary Fig. 1d and Supplementary Dataset S1d).

To dissociate the effect of food training on the acquisition of cocaine self-administration, a separate cohort of male mice that were not food trained was assessed. In this non-food-trained cohort, WT and ChI-D2RKO mice progressively acquired a significant number of cocaine infusions (0.3 mg/kg per infusion) across daily sessions (two-way repeated measures ANOVA, session: $F_{(3.473,38.20)} = 15.18$, $P < 0.0001$; genotype: $F_{(1,11)} = 0.0007525$, $P = 0.9786$; session × genotype: $F_{(16,176)} = 0.6718$, $P = 0.8190$) (Fig. 1a and Supplementary Dataset 1a). Moreover, both WT and ChI-D2RKO mice were able to differentiate between active and inactive levers for cocaine (three-way repeated measures ANOVA, session: $F_{(16,352)} = 6.128$, $P < 0.0001$; genotype: $F_{(1,22)} = 0.1030$, $P = 0.7513$; lever: $F_{(1,22)} = 31.06$, $P < 0.0001$; session × genotype: $F_{(16,352)} = 0.3524$, $P = 0.9910$; session × lever: $F_{(16,352)} = 6.202$, $P < 0.0001$; genotype × lever: $F_{(1,22)} = 0.05036$, $P = 0.8245$; session × genotype × lever: $F_{(16,352)} = 0.2427$, $P = 0.9990$) (Fig. 1b and Supplementary Dataset 1b). These findings indicate that prior operant food training had no effect on the animals' level of responding for cocaine when comparing between genotypes. Next, all subjects underwent assessment using the progressive ratio (PR) schedule of reinforcement to evaluate motivation under an increasing escalation of response requirements to obtain each subsequent cocaine infusion. Interestingly, under the PR schedule, ChI-D2RKO mice earned significantly fewer cocaine infusions, resulting in a lower breakpoint compared to WT controls ($P = 0.0065$) (Fig. 1c and Supplementary Dataset 1c). Similarly, ChI-D2RKO mice exhibited fewer active lever presses than WT mice ($P = 0.0174$) (Fig. 1d and Supplementary Dataset 1d). These results suggest that male ChI-D2RKO mice perceive cocaine as less valuable compared to their WT littermates. This conclusion is supported by a significant decrease in motivation to obtain cocaine infusions under the PR schedule observed in ChI-D2RKO mice.

No differences in the number of saline infusions earned (two-way repeated measures ANOVA, session: $F_{(2.585,18.10)} = 3.398$, $P = 0.0458$; genotype: $F_{(1,7)} = 0.09446$, $P = 0.7675$; session × genotype: $F_{(11,77)} = 0.6744$, $P = 0.7583$) (Supplementary Fig. S2a and

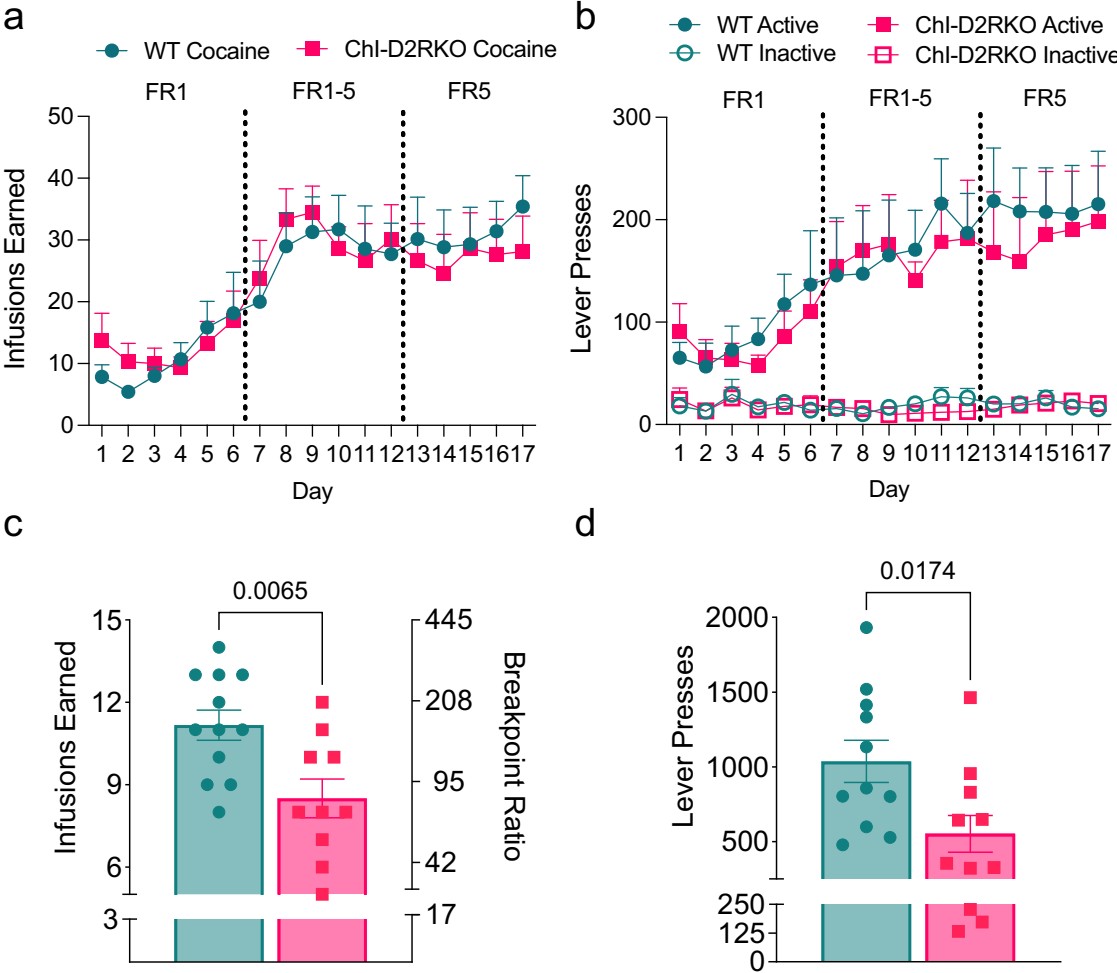

**Fig. 1 | Motivation to obtain cocaine is significantly reduced in ChI-D2RKO mice. a** Number of cocaine infusions earned and **b** lever presses during cocaine intravenous self-administration (0.3 mg/kg/infusion) in non-food trained WT (N = 7) and ChI-D2RKO (N = 6) mice during an FR5TO20 second schedule of reinforcement. **a** Two-way repeated measures ANOVA, session: $F_{(3.473,38.20)} = 15.18$, P < 0.0001; genotype: $F_{(1,11)} = 0.0007525$, P = 0.9786; session × genotype: $F_{(16,176)} = 0.6718$, P = 0.8190 and **b** three-way repeated measures ANOVA, session: $F_{(16,352)} = 6.128$, P < 0.0001; genotype: $F_{(1,22)} = 0.1030$, P = 0.7513; lever:

$F_{(1,22)} = 31.06$, P < 0.0001; session × genotype: $F_{(16,352)} = 0.3524$, P = 0.9910; session × lever: $F_{(16,352)} = 6.202$, P < 0.0001; genotype × lever: $F_{(1,22)} = 0.05036$, P = 0.8245; session × genotype × lever: $F_{(16,352)} = 0.2427$, P = 0.9990. **c** Number of cocaine infusions (0.3 mg/kg/infusion) earned and **d** lever presses during a progressive ratio schedule of reinforcement in food-trained (N = 4 WT, N = 5 ChI-D2RKO) and non-food trained (N = 7 WT, N = 6 ChI-D2RKO) mice. **c** Unpaired two-tailed t-test: P = 0.0168, t = 2.608, df = 20 and **d** unpaired two-tailed t-test: P = 0.0174, t = 2.593, df = 20. Values shown are mean ± SEM.

Supplementary Dataset S2a) nor active/inactive lever pressing (three-way repeated measures ANOVA, session: $F_{(1.883,13.18)} = 2.655$, P = 0.1096; genotype: $F_{(1,7)} = 0.2406$, P = 0.6388; lever: $F_{(1,7)} = 24.54$, P = 0.0016; session × genotype: $F_{(11,77)} = 0.6934$, P = 0.7408; session × lever: $F_{(2.876,20.13)} = 2.305$, P = 0.1095; genotype × lever: $F_{(1,7)} = 0.1017$, p = 0.7591; session × genotype × lever: $F_{(11,77)} = 0.6888$, P = 0.7451) (Supplementary Fig. S2b and Supplementary Dataset S2b) during acclimation were observed. Additionally, in the saline PR condition, WT and ChI-D2RKO mice exhibited no differences in the number of infusions earned (P = 0.7746) (Supplementary Fig. S2c and Supplementary Dataset S2c) or in active lever pressing (P = 0.8005) (Fig. S2d).

### Loss of D2R signaling in ChIs causes severe alterations of the nucleus accumbens transcriptome

Previous studies using ChI-D2RKO mice showed an altered transcriptomic profile in the dorsomedial striatum in response to contingent cocaine administration[18], as exemplified by the lack of induction of critical cocaine-induced immediate early genes (IEGs) in the mutants. These results suggested that these alterations of gene expression observed after the first exposure to cocaine can possibly

evolve into long-term changes underlying the reduced PR breakpoint of ChI-D2RKO mice.

To investigate this point, we performed RNA-sequencing (RNA-seq) analyses of the NAcc from male WT and ChI-D2RKO mice collected immediately after the end of the PR schedule for each condition. Analyses of differentially expressed genes (DEGs) between WT and ChI-D2RKO mice identified >2000 transcripts when comparing saline vs cocaine IVSA (P < 0.05) (Fig. 2a). Using $\log_2$ fold change cut-offs of ±0.3, we found 897 DEGs downregulated and 1398 upregulated in response to cocaine in the WT as compared to ChI-D2RKO NAcc (P < 0.05) (Fig. 2b). Gene ontology (GO) analyses showed alterations in genes related to learning and memory, synaptic plasticity, signaling, and transcription regulation (Fig. 2c).

Importantly, we found that several genes previously associated with SUDs[31] showed increased expression in WT but not in ChI-D2RKO mice, among these, IEGs such as *Fos*, *Nr4a1*, *Egr1*, and *Fosb*[31]. Promoters of genes in these categories, like Fosb, have previously been shown to exhibit enriched histone acetylation in response to cocaine exposure[31] and shown to be induced mostly in D1R⁺ MSNs[32–35]. Notably, our data revealed a significant interaction between genotype and cocaine

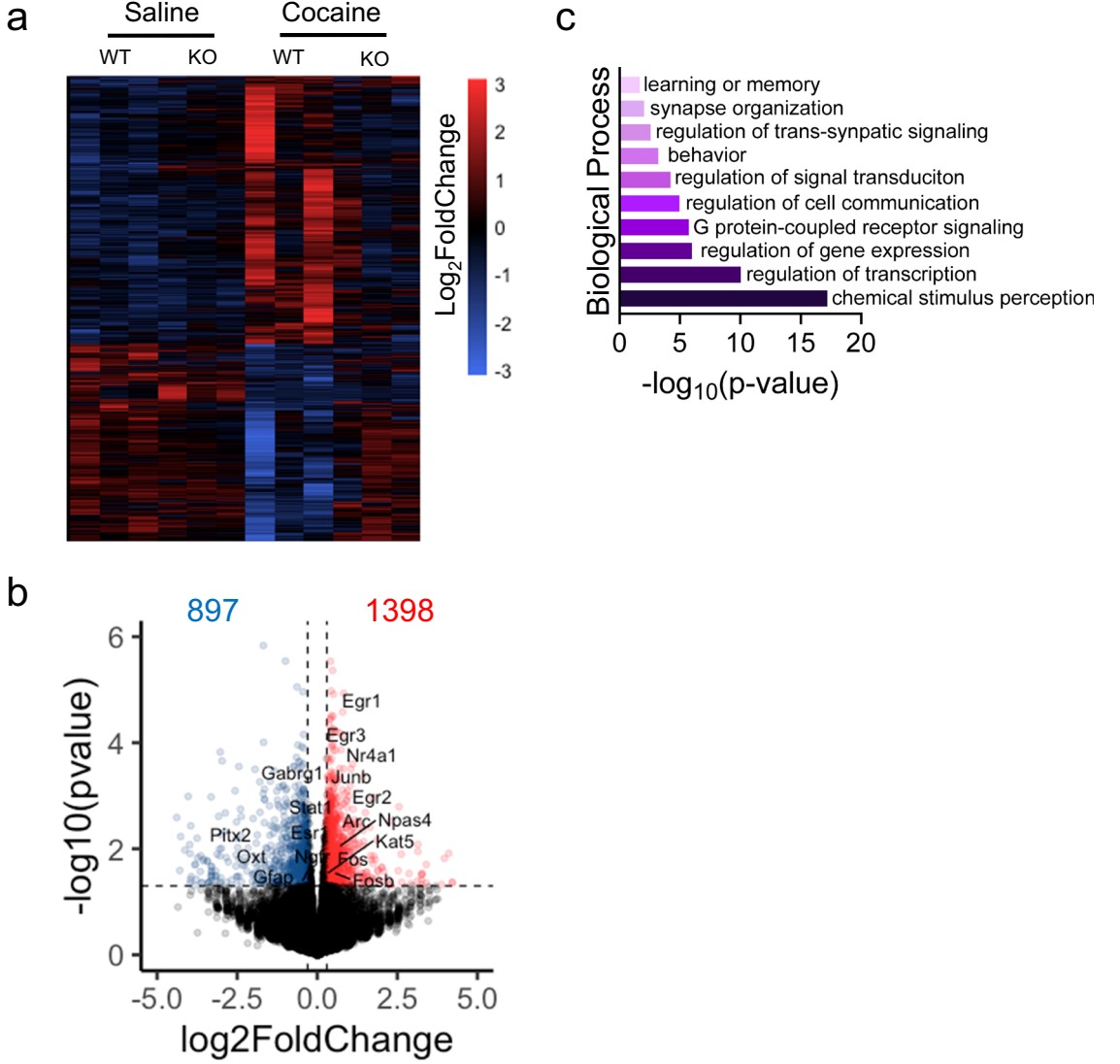

**Fig. 2 | Transcriptomic analysis of the NAcc in WT and ChI-D2RKO following PR. a** Heatmap illustrating the $\log_2$ fold change of genes differentially expressed ($P < 0.05$) in saline or cocaine self-administering WT and ChI-D2RKO mice (N = 3/group); upregulation (red), downregulation (blue). **b** Volcano plot based on the $-\log10$ p-value vs the $\text{Log}_2$ Fold Change of the NAcc transcriptome (WT cocaine vs ChI-D2RKO cocaine). Cut-offs are shown as dotted lines ($P < 0.05$) and $\log_2$FC ± 0.3. **c** PANTHER Gene Ontology based on differentially expressed genes from (**b**). **a–c** Differential expression was analyzed with DESeq2 using a negative binomial model and Wald test; P-values were FDR-adjusted (Benjamini–Hochberg).

treatment, along with the marked increase in Chrm4 (M4R) gene expression in WT (P = 0.0091) but not ChI-D2RKO (P = 0.8154) mice (Supplementary Fig. S3 and Supplementary Dataset S3).

We also compared gene expression between cocaine only treated groups to identify genotype-dependent differences specifically in the cocaine condition. Using the same cut-offs as for the analysis shown in Fig. 2, we found 1008 genes downregulated and 1074 genes upregulated (WT cocaine vs ChI-D2RKO cocaine; P < 0.05) (Supplementary Fig. S4a and Supplementary Dataset S4a), also involved in nervous system development, transcription, and synaptic transmission (Supplementary Fig. S4b and Supplementary Dataset S4b). Importantly, M4R and Kat5 emerged as significant DEGs from this comparison, confirming the differential cocaine-mediated effects on their expression between genotypes.

### Activation of M4R reduces the effects of cocaine in male WT mice

The induced expression of M4R in WT mice led us to hypothesize that the DA and D2R-mediated control of ACh and its signaling critically

regulate cocaine intake and may account for the diminished responses of ChI-D2RKO mice in PR. Thus, we examined the behavioral effects of the M4R positive allosteric modulator (PAM) VU0467154 when administered in combination with cocaine in male WT mice.

Administration of VU0467154 (5 mg/kg i.p.) had no impact on baseline motor ability (P > 0.9999) or coordination (P = 0.1023) in male WT mice (Fig. 3a, b). However, when given 15 min before cocaine, VU0467154 significantly reduced cocaine-dependent motor activity (P = 0.0014) (Fig. 3a, c), consistent with previously published findings[36].

Next, we measured the effect of VU0467154 on the rewarding properties of cocaine by the CPP test. Gross motor activity during the pre-conditioning (P = 0.9820) (Supplementary Fig. S5a and Supplementary Dataset S5a) or test (P = 0.3985) (Supplementary Fig. S5b and Supplementary Dataset S5b) sessions did not differ between treatment conditions. Interestingly, VU0467154 alone did not elicit either rewarding or aversive effects, as CPP scores in VU0467154/saline-treated mice were comparable to those in the vehicle/saline control group (P = 0.9998) (Fig. 3d). While vehicle/cocaine treated WT mice

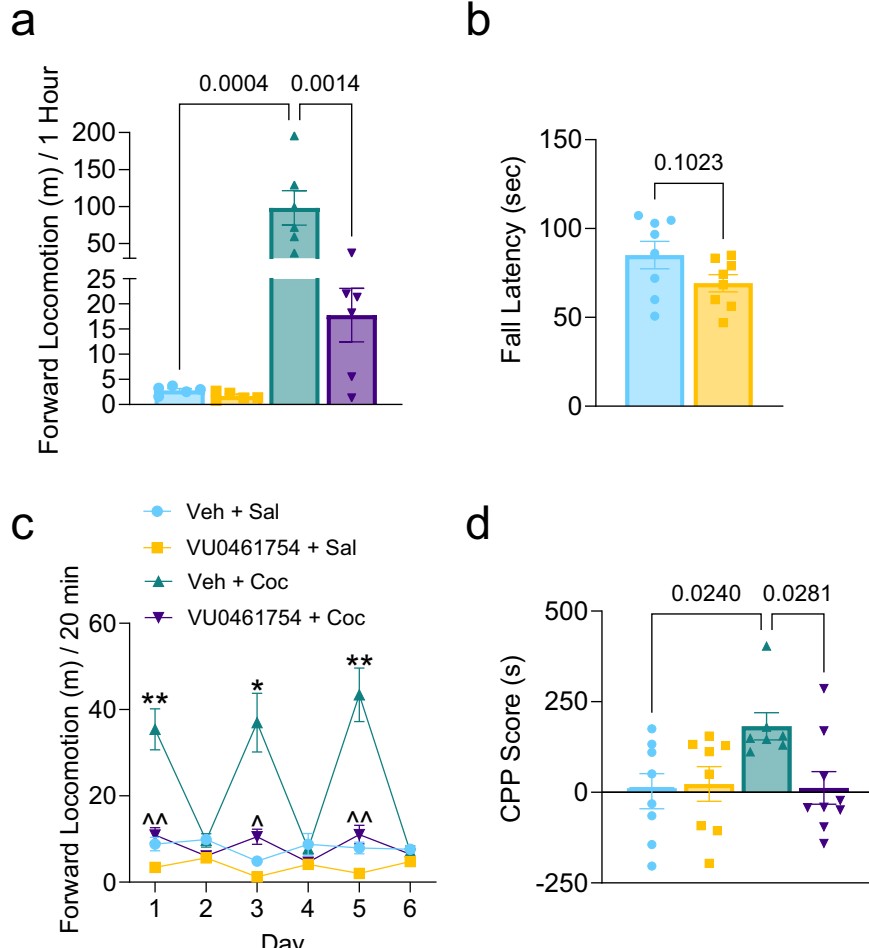

**Fig. 3 | The M4R positive allosteric modulator VU04567154 lowers cocaine's reinforcing properties without impairing motor control. a** Forward locomotion in WT mice treated with vehicle or VU0467154 (5 mg/kg; i.p.) 15 min prior to saline or cocaine. Activity was recorded for 1-h following saline or cocaine administration (N = 5–6/group). One-Way ANOVA: $F_{(3,18)}$ = 12.65, P = 0.0001. Bonferroni post-hoc test, vehicle/saline vs vehicle/cocaine: P = 0.0004; vehicle/cocaine vs VU0467154/cocaine: P = 0.0014. **b** Average fall latency of three trials on the rotarod test set at 24 RPM 15 min after being administered by vehicle or VU0467154 (N = 8/group). Unpaired two-tailed t-test: P = 0.1023, t = 1.748, df = 14. **c** Forward locomotor activity during the conditioning phase of the conditioned place preference (CPP)

protocol. Two-way repeated measures ANOVA, day: $F_{(2.085,56.30)}$ = 31.37, P < 0.0001; treatment: $F_{(3,27)}$ = 21.88, P < 0.0001; day × treatment: $F_{(15,135)}$ = 27.79, P < 0.0001. Bonferroni post-hoc test, vehicle/saline vs vehicle/cocaine: day 1 **P = 0.0057, day 3 *P = 0.0183, day 5 **P = 0.064; vehicle/cocaine vs VU0467154/cocaine: day 1 ^^P = 0.0087, day 3 ^P = 0.0440, day 5 ^^P = 0.0088. **d** CPP score after conditioning with either saline/saline, VU0467154/saline, vehicle/cocaine, or VU0467154/cocaine in WT mice (N = 7–9/group). One-way ANOVA: $F_{(3,28)}$ = 3.182, P = 0.0392. Bonferroni post-hoc test, vehicle/saline vs vehicle/cocaine: P = 0.0240; vehicle/cocaine vs VU0467154/cocaine: P = 0.0281. Values shown are mean ± SEM.

showed a significant increase in CPP score compared to vehicle/saline-treated mice (P = 0.0240), those pretreated with VU0467154 prior to cocaine failed to exhibit CPP, differing significantly from the cocaine only group (P = 0.0281) (Fig. 3d).

To evaluate whether the decreased motivation of ChI-D2RKO during the PR schedule could arise from an increased M4R stimulation, we trained male WT mice to respond to cocaine under an FR5TO20 schedule and administered VU0467154 the day of the test 15 min before the PR session. Importantly, we found no significant difference in cocaine infusions earned (two-way repeated measures ANOVA, session: $F_{(2.724,32.01)}$ = 2.112, P = 0.1231; treatment: $F_{(1,12)}$ = 0.007259, P = 0.9335; session × treatment: $F_{(4,47)}$ = 0.5274, P = 0.7161) (Fig. 4a) or active lever pressing (three-way repeated measures ANOVA, session: $F_{(4,60)}$ = 0.1519, P = 0.9614; treatment: $F_{(1,12)}$ = 0.4331, P = 0.5130; lever: $F_{(1,60)}$ = 50.99, P < 0.0001; session × treatment: $F_{(4,60)}$ = 0.03793, P = 0.9972; session × lever: $F_{(4,60)}$ = 0.06045, P = 0.9931; treatment × lever: $F_{(1,60)}$ = 0.9159, P = 0.3424; session × treatment × lever: $F_{(4,60)}$ = 0.1821, P = 0.9468) (Fig. 4b) between mice assigned to receive either vehicle or VU0467154 for the PR session. Therefore, following the 5-day stabilization period,

mice were given either vehicle or VU0467154 (5 mg/kg) 15 min before the PR session accordingly. Remarkably, during this test, VU0467154-treated mice exhibited a significant reduction in cocaine rewards earned (P = 0.0339) (Fig. 4c) and lever presses (P = 0.0314) (Fig. 4d) compared to vehicle-treated mice. These findings suggest that the absence of D2R in ChIs enhances ACh-dependent M4R signaling, ultimately reducing the motivational effects of cocaine.

The effect of M4R signaling in cocaine CPP and IVSA, as well as in the behavioral and cellular responses to both acute and chronic cocaine[18], raised the question of the D2R-dependent control of ACh and implication of M4R expression in controlling responses to cocaine. Importantly, RNA-seq data showed a cocaine-mediated induction of M4R mRNA expression in the NAcc of WT mice, which is absent in ChI-D2RKO mice (Supplementary Fig. S3). In support of this data, we performed fluorescent in situ hybridization (FISH) of male WT and ChI-D2RKO mice chronically administered (once daily for 7 days) saline or cocaine (15 mg/kg i.p.). FISH analyses on fixed coronal cryostat sections were made using probes specific for M4R, D1R, and D2R; thus allowing for the identification of the MSN subtype(s) subject to cocaine-induced

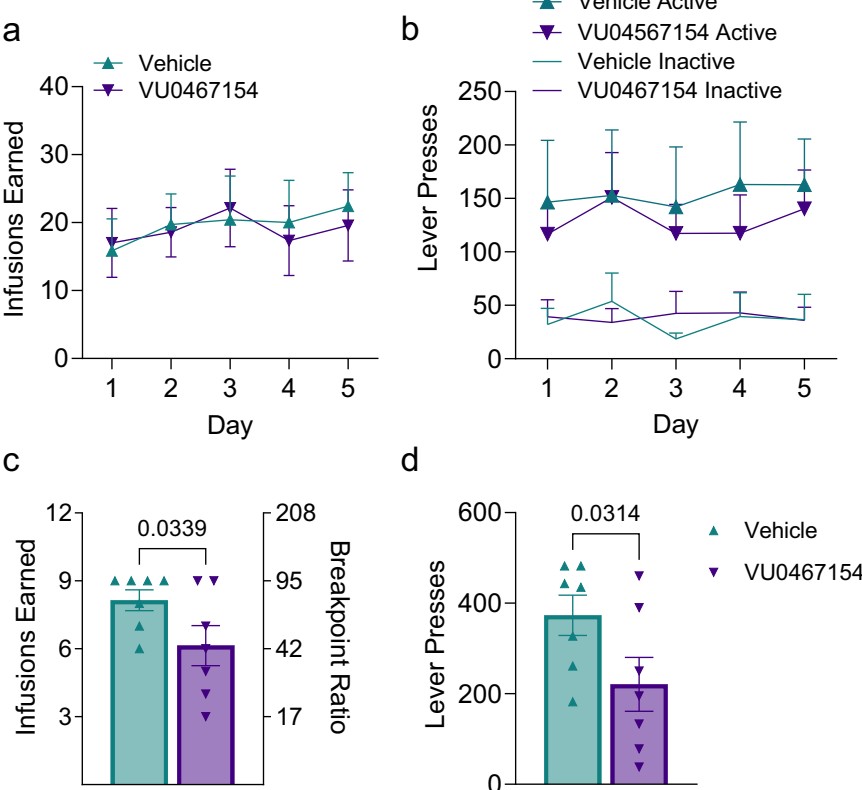

**Fig. 4 | M4R positive allosteric modulation lowers cocaine's reinforcing properties in WT mice. a** Cocaine infusions earned and **b** active lever pressing in mice during 5 days of stabilized cocaine responding in animals that were assigned to receive either vehicle or VU0467154 for the PR test (**c, d**) (N = 7/group). **a** Two-way repeated measures ANOVA, session: $F_{(2.724, 32.01)} = 2.112$, P = 0.1231; treatment: $F_{(1,12)} = 0.007259$, P = 0.9335; session × treatment: $F_{(4,47)} = 0.5274$, P = 0.7161 and **b** three-way repeated measures ANOVA, session: $F_{(4,60)} = 0.1519$, P = 0.9614; treatment: $F_{(1,60)} = 0.4331$, P = 0.5130; lever: $F_{(1,60)} = 50.99$, P < 0.0001; session × treatment: $F_{(4,60)} = 0.03793$, P = 0.9972; session × lever: $F_{(4,60)} = 0.06045$, P = 0.9931; treatment × lever: $F_{(1,60)} = 0.9159$, P = 0.3424; session × treatment × lever: $F_{(4,60)} = 0.1821$, P = 0.9468. **c** Number of cocaine infusions (0.3 mg/kg/infusion) earned and **d** active lever presses during the progressive ratio schedule of reinforcement when given vehicle or VU0467154 (5 mg/kg; i.p.) 15 min prior to the start of the session (N = 7/group). **c** Unpaired one-tailed t-test: P = 0.0339 t = 2.007, df = 12 and **d** unpaired one-tailed t-test: P = 0.0314, t = 2.051, df = 12. Values shown are mean ± SEM.

M4R upregulation (Fig. 5a, Supplementary Fig. S6a and Supplementary Dataset S6a). Quantifications of gene expression were performed on confocal images by measuring fluorescent intensity of M4R, D1R, and D2R-specific puncta in $D1R^+$ and $D2R^+$ MSNs. We observed a cocaine-induced upregulation of D1R in the NAcc of WT mice (P = 0.0020), absent in the ChI-D2RKO (P = 0.7178) (Fig. 5b), while D2R expression was not significantly altered in the NAcc of either genotype in response to cocaine (WT: P = 0.1531, ChI-D2RKO: P = 0.6383) (Fig. 5d).

In line with our RNA-seq results, we also found a striking increase in M4R expression in the NAcc of WT but not ChI-D2RKO mice following chronic cocaine treatment. Importantly, this upregulation of M4R was exclusively detected in $D1R^+$ MSNs (WT: P = 0.0011, ChI-D2RKO: P = 0.4277) (Fig. 5c), whereas no difference in M4R expression was observed between genotype or treatment in $D2R^+$ MSNs (WT: P = 0.8263, ChI-D2RKO: P = 0.9542) (Fig. 5e). Notably, the increase in M4R expression following contingent chronic cocaine treatment was not present in the dorsolateral striatum (DLS; Fig. S6) of $D1R^+$ MSNs of WT mice (P = 0.1533) (Supplementary Fig. S6b and Supplementary Dataset S6b). In the ChI-D2RKO DLS, we even found a decrease of M4R expression in $D1R^+$ MSNs (P = 0.0009) of cocaine treated mice as compared to the saline (Supplementary Fig. S6b). No differences between treatments were observed in $D2R^+$ MSNs (WT: P = 0.6108, ChI-D2RKO: P = 0.9540) in the DLS of both genotypes (Supplementary Fig. S6c and Supplementary Dataset S6c). These results strongly suggest a NAcc-specific mechanism by which ACh signaling regulates cocaine responses through modulation of $D1R^+$ MSNs.

## The acetyltransferase Kat5 participates in striatal cocaine effects

Importantly, RNA-seq analysis (Fig. 2b, Supplementary Fig. S4a and Supplementary Dataset S4a) identified lysine acetyltransferase 5 (Kat5) as a novel component of cocaine-induced genes in the NAcc of WT mice differentially expressed among genotypes. Kat5 is a histone acetyltransferase (HAT), suggesting a mechanism by which cocaine's effects depend on a strict control of DA and D2R signaling in ChIs. This control, absent in ChI-D2RKO mice, is required to induce epigenetic modifications that contribute to cocaine's outcome on striatal circuits.

Previously, we demonstrated that tropicamide, a muscarinic receptor antagonist with high selectivity for M4R, successfully reversed the behavioral and cellular phenotypes observed in ChI-D2RKO mice following contingent cocaine exposure, restoring them to WT levels[18]. Based on these findings, we treated male mice of both genotypes with either vehicle (saline) or tropicamide (10 mg/kg i.p.) 15 min before cocaine and analyzed Kat5 induction in NAcc extracts collected 1 h after treatment (Fig. 6). Interestingly, western blot analysis revealed that the induction of Kat5 by cocaine IVSA in WT mice is not restricted to the long-term effects of the drug, as we observed a significant increase in the Kat5 protein expression after contingent cocaine administration (P = 0.0176), which is absent in the mutants (P > 0.9999) (Fig. 6a). Tropicamide administration prior to cocaine restored Kat5 induction in ChI-D2RKO mice as compared to those receiving treatment with vehicle/cocaine (P = 0.0099), bringing its levels to that of saline/cocaine treated WT animals (Fig. 6a).

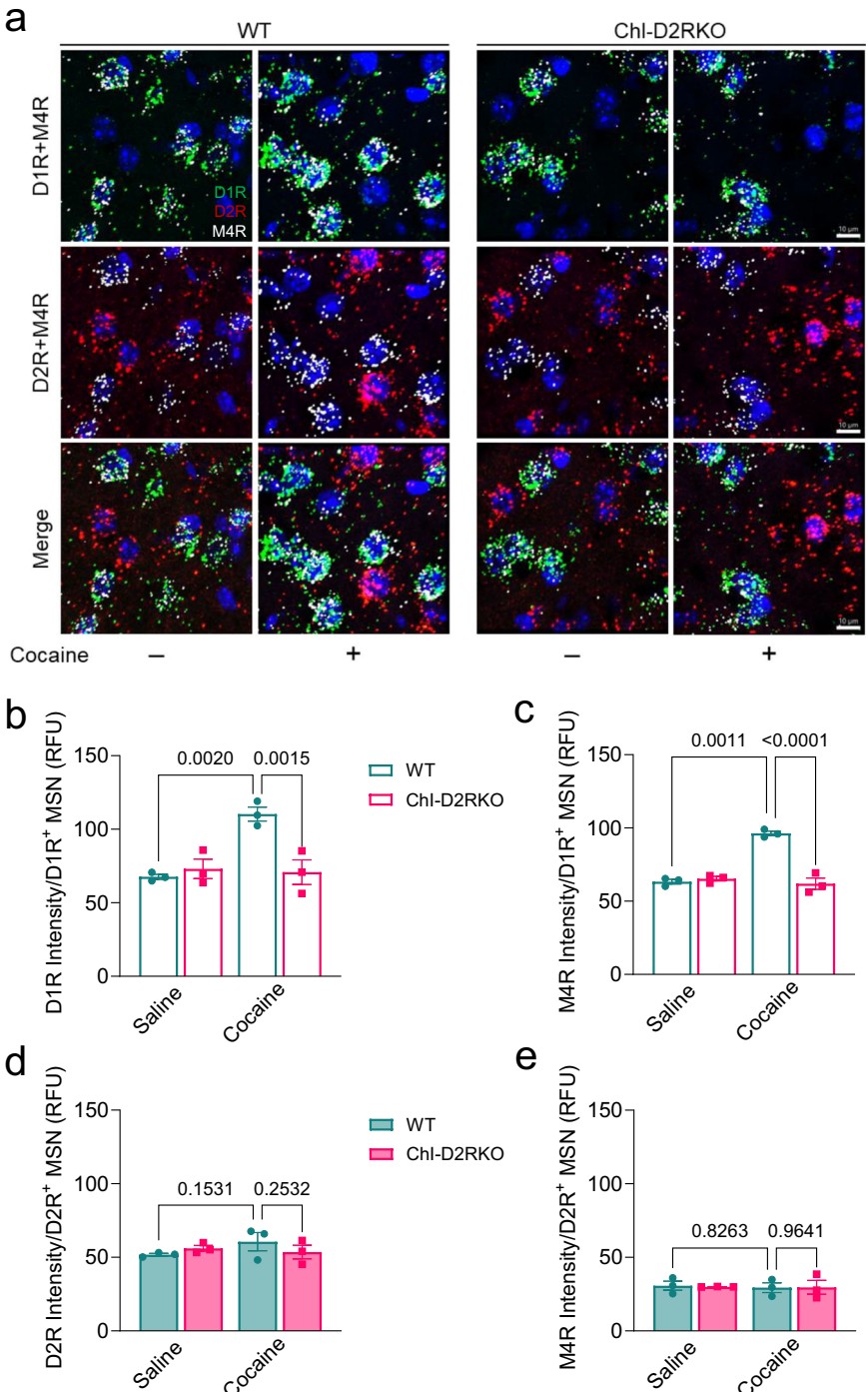

**Fig. 5 | Cocaine upregulates expression of D1R and M4R in D1R+ MSNs.**
**a** Representative images of FISH experiments showing D1R (green), D2R (red), and M4R (white) expression in the NAcc of WT and ChI-D2RKO mice (N = 3/group) treated with cocaine (15 mg/kg i.p.) chronically for 7 days (blue: DAPI). Scale bar: 10 μm. **b** Quantification of the fluorescent intensity (RFU) of D1R per D1R+ MSN in the NAcc. Two-way ANOVA, genotype: $F_{(1,4)}$ = 5.593, P = 0.0773; treatment: $F_{(1,4)}$ = 23.14, P = 0.0086; genotype × treatment: $F_{(1,4)}$ = 28.72, P = 0.0059. Tukey post-hoc test, WT saline vs WT cocaine: P = 0.0020; WT cocaine vs ChI-D2RKO cocaine: P = 0.0015. **c** Quantification of the fluorescent intensity (RFU) of M4R per D1R+ MSN in the NAcc. Two-way ANOVA, genotype: $F_{(1,4)}$ = 71.99, P = 0.0011; treatment: $F_{(1,4)}$ = 28.93, P = 0.0058; genotype × treatment: $F_{(1,4)}$ = 43.90, P = 0.0027.

Tukey post-hoc test: WT saline vs WT cocaine: P = 0.0011; WT cocaine vs ChI-D2RKO cocaine: P < 0.0001. **d** Quantification of the fluorescent intensity of D2R per D2R+ MSN in the NAcc. Two-way ANOVA, genotype: $F_{(1,4)}$ = 0.09096, P = 0.7780; treatment: $F_{(1,4)}$ = 0.7844, P = 0.4258; genotype × treatment: $F_{(1,4)}$ = 2.572, P = 0.1840. Tukey post-hoc test, WT saline vs WT cocaine: P = 0.1531; WT cocaine vs ChI-D2RKO cocaine: P = 0.2532. **e** Quantification of the fluorescent intensity of M4R per D2R+ MSN in the NAcc. Two-way ANOVA, genotype: $F_{(1,4)}$ = 0.01364, P = 0.9126; treatment: $F_{(1,4)}$ = 0.04360, P = 0.8448; genotype × treatment: $F_{(1,4)}$ = 0.01497, P = 0.9085. Tukey post-hoc test, WT saline vs WT cocaine: P = 0.8263; WT cocaine vs ChI-D2RKO cocaine: P = 0.9641. Values shown are mean ± SEM.

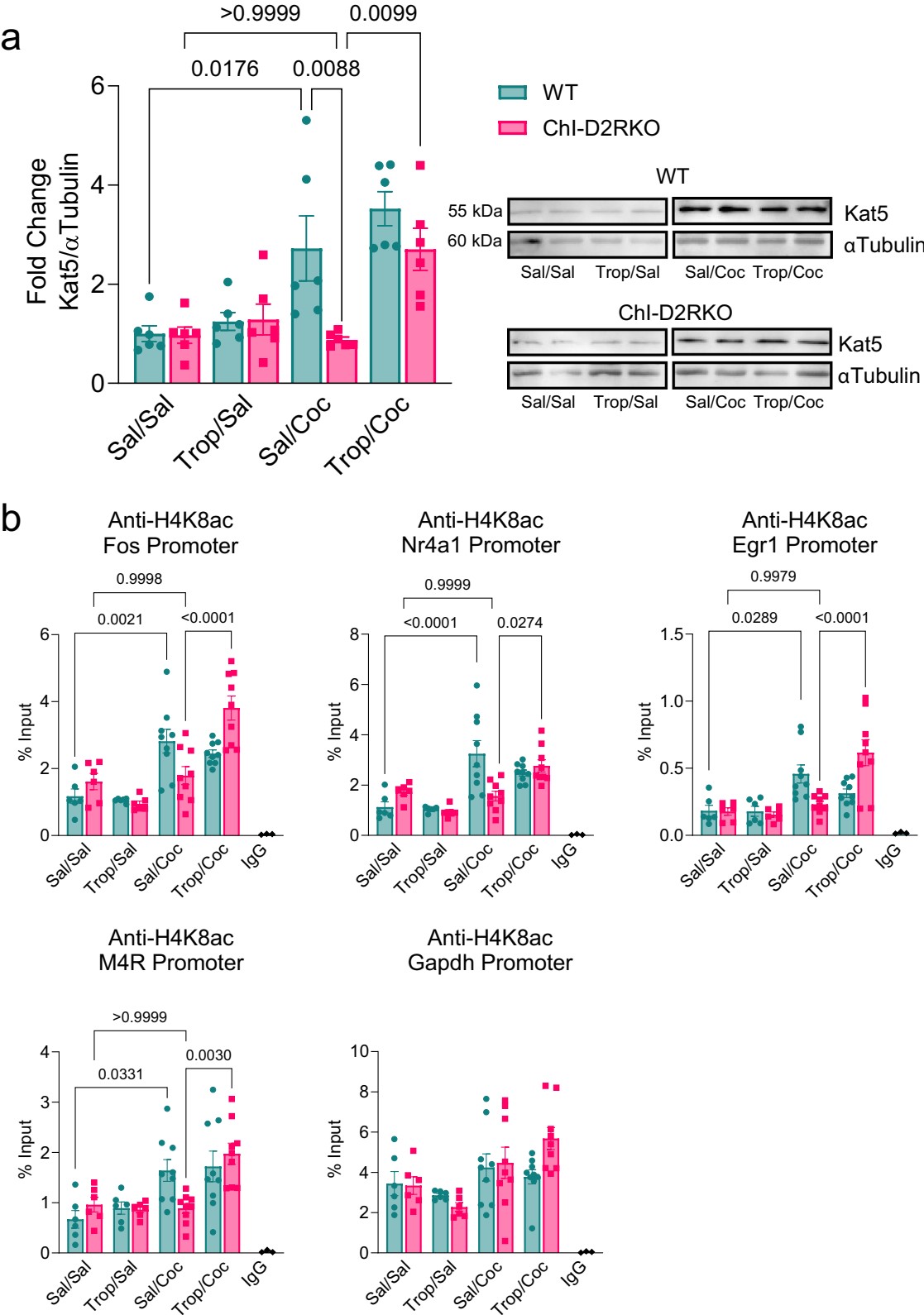

Kat5 acetylates histone H4 on lysine 8 (H4K8ac)[21,37], and H4K8ac is enriched at the promoters of IEGs upon cocaine treatment[31]. Thus, we collected NAcc extracts from male WT and ChI-D2RKO mice 1 h after acute treatment with saline or tropicamide (10 mg/kg i.p.) 15 min prior to cocaine and performed chromatin immunoprecipitation (ChIP). H4K8ac enrichment was tested at the promoters of IEGs previously mentioned and M4R, and we found a significant enrichment in the WT extracts from mice treated with cocaine vs saline (WT saline/saline vs saline/cocaine; Fos: P = 0.0021, Nr4a1: P < 0.0001, Egr1: P = 0.0289, M4R: P = 0.0331) (Fig. 6b). H4K8ac enrichment was absent on the same promoters in the ChI-D2RKO extracts (ChI-D2RKO saline/saline vs saline/cocaine; Fos: P = 0.9998, Nr4a1: P = 0.9999, Egr1: P = 0.9979,

**Fig. 6 | M4R antagonism in ChI-D2RKO mice restores cocaine-induced gene expression. a** *Left:* Bar graph showing fold change in expression of Kat5/αTubulin by Western blot in NAcc extracts 1-h after acute treatment with vehicle or tropicamide (10 mg/kg i.p.) 15 min prior to saline or cocaine (15 mg/kg i.p.). Fold changes shown are relative to WT saline (N = 6/group). Two-way ANOVA, genotype: $F_{(1,40)} = 7.721$, P = 0.0083; treatment: $F_{(3,40)} = 15.59$, P < 0.0001; genotype × treatment: $F_{(3,40)} = 3.381$, P = 0.0274. Tukey post-hoc test, WT sal/sal vs WT sal/coc: P = 0.0176; ChI-D2RKO sal/sal vs ChI-D2RKO sal/coc: P > 0.9999; WT sal/coc vs ChI-D2RKO sal/coc: P = 0.0088. *Right:* Representative Western Blot images of samples from *Left*. **b** Bar graphs of H4K8ac enrichment of Fos, Nr4a1, Egr1, M4R, and Gapdh promoters in WT and ChI-D2RKO mice treated with saline or cocaine in the presence or absence of tropicamide, as indicated, shown as a percentage of input (N = 6–9/group). Two-way ANOVA, Fos) genotype: $F_{(1,52)} = 0.7380$, P = 0.3942; treatment: $F_{(3,52)} = 25.11$, P < 0.0001; genotype × treatment: $F_{(3,52)} = 8.256$, P = 0.0001. Nr4a1) genotype: $F_{(1,52)} = 1.293$, P = 0.2607; treatment: $F_{(3,52)} = 15.45$, P < 0.0001; genotype × treatment: $F_{(3,52)} = 7.506$, P = 0.0003. Egr1) genotype:

$F_{(1,52)} = 0.08207$, P = 0.7756; treatment: $F_{(3,52)} = 12.51$, P < 0.0001; genotype × treatment: $F_{(3,52)} = 8.934$, P < 0.0001. M4R) genotype: $F_{(1,52)} = 0.1666$, P = 0.6849; treatment: $F_{(3,52)} = 11.19$, P < 0.0001; genotype × treatment: $F_{(3,52)} = 3.174$, P = 0.0317. Gapdh) genotype: $F_{(1,52)} = 0.7933$, P = 0.3772; treatment: $F_{(3,52)} = 5.645$, P = 0.0020; genotype × treatment: $F_{(3,52)} = 1.921$, P = 0.1377. Tukey post-hoc test, Fos) WT sal/sal vs WT sal/coc: P = 0.0021; ChI-D2RKO sal/sal vs ChI-D2RKO sal/coc: P > 0.9999; ChI-D2RKO sal/coc vs ChI-D2RKO trop/coc: P < 0.0001; Nr4a1) WT sal/sal vs WT sal/coc: P < 0.0001; ChI-D2RKO sal/sal vs ChI-D2RKO sal/coc: P > 0.9999; ChI-D2RKO sal/coc vs ChI-D2RKO trop/coc: P = 0.0274; Egr1) WT sal/sal vs WT sal/coc: P = 0.0289; ChI-D2RKO sal/sal vs ChI-D2RKO sal/coc: P = 0.9970; ChI-D2RKO sal/coc vs ChI-D2RKO trop/coc: P < 0.0001; M4R) WT sal/sal vs WT sal/coc: P = 0.0331; ChI-D2RKO sal/sal vs ChI-D2RKO sal/coc: P > 0.9999; ChI-D2RKO sal/coc vs ChI-D2RKO trop/coc: P = 0.0030; Gapdh) WT sal/sal vs WT sal/coc: P = 0.9745; ChI-D2RKO sal/sal vs ChI-D2RKO sal/coc: P = 0.8600; ChI-D2RKO sal/coc vs ChI-D2RKO trop/coc: P = 0.7156. Values shown are mean ± SEM.

M4R: P > 0.9999) (Fig. 6b). Importantly, when ChI-D2RKO mice were given tropicamide prior to cocaine, a robust enrichment of H4K8ac was restored at all the tested promoters (ChI-D2RKO saline/cocaine vs tropicamide/cocaine: Fos (P < 0.0001), Nr4a1 (P = 0.0274), Egr1 (P < 0.0001), and M4R (P = 0.0030)) (Fig. 6b). Tropicamide did not have significant effects in WT controls (WT saline/cocaine vs tropicamide/cocaine; Fos: P = 0.9601, Nr4a1: P = 0.3855, Egr1: P = 0.5046, M4R: P > 0.9999) (Fig. 6b). Across all conditions, the Gapdh promoter, used as a positive control, showed no genotype- or treatment-dependent differences (two-way ANOVA, genotype: $F_{(1,52)} = 0.7933$, P = 0.3772; treatment: $F_{(3,52)} = 5.645$, P = 0.0020; genotype × treatment: $F_{(3,52)} = 1.921$, P = 0.1377) (Fig. 6b). Thus, the absence of D2R signaling in ChIs leads to major changes in ACh-mediated signaling in the striatum, contributing to inhibition of cocaine-dependent Kat5 induction and of the enrichment of H4K8ac at target gene promoters. These events can be restored in mice lacking inhibitory D2R-dependent control of ChIs by blocking M4R signaling, strongly implicating DA and D2R-mediated inhibition of ACh signaling in the Kat5-dependent modulation of acetylation of H4K8 in the striatal response to cocaine.

## Kat5 inhibition prevents the acute and chronic effects of cocaine

To further support the critical role of Kat5 in the response to cocaine, we investigated whether blocking Kat5 activity would affect the behavioral and cellular effects of the drug. Thus, male WT and ChI-D2RKO mice were administered vehicle (saline) or NU9056 (2.5 mg/kg i.p.), a well-characterized Kat5 inhibitor[38], 1 h before cocaine (15 mg/kg i.p.) (Fig. 7, Supplementary Fig. S7 and Supplementary Dataset S7), and motor behavior was scored for 1 h. Two conditions were tested, after both acute administration (Supplementary Fig. S7 and Supplementary Dataset S7a) and a contingent chronic cocaine regimen (once daily for 7 days; Fig. 7a). Strikingly, in WT mice NU9056 suppressed cocaine's psychomotor effects in both acute (WT saline/cocaine vs NU9056/cocaine; P < 0.0001) (Supplementary Fig. S7 and Supplementary Dataset S7a) and chronic conditions (WT saline/cocaine vs NU9056/cocaine; P < 0.0001) (Fig. 7a) in a manner comparable to that of cocaine treated ChI-D2RKO animals. Western blot analysis of NAcc extracts obtained from WT mice 1 h after the final administration of chronic NU9056/cocaine treatment confirmed both cocaine induction of Kat5 (P = 0.0109) and its inhibition by NU9056 (P = 0.0025) (Fig. 7b). These findings support the critical role of Kat5 in cocaine-mediated events.

Given that Kat5 is expressed by most cell types[22,39] in the brain, we sought to determine the MSN subtype(s) having the largest induction of Kat5 in the striatum of animals of both genotypes following the 7-day chronic cocaine regimen in the presence or absence of NU9056. For this, we performed FISH on coronal striatal cryostat sections obtained from these animals 1 h after the final day of treatment using probes targeting Kat5, D1R, and D2R (Fig. 7c), and quantified the

number of Kat5 puncta localized in D1R+ (Fig. 7d) and D2R+ (Fig. 7e) MSNs in the NAcc. Importantly, we found that D1R+ MSNs are the site of the largest induction of Kat5 in response to cocaine treatment in WT mice (WT saline/saline vs saline/cocaine; P = 0.0191) (Fig. 7d). In D2R+ MSNs, only a trend toward Kat5 induction by cocaine was observed in the WT (P = 0.1881), with notably fewer puncta per cell compared to D1R+ MSNs (Fig. 7e). In ChI-D2RKO mice, Kat5 induction in response to contingent chronic cocaine treatment was absent in both MSN subtypes (ChI-D2RKO saline/saline vs saline/cocaine; D1R+ MSNs: P = 0.1578, D2R+ MSNs: P = 0.0610) (Fig. 7d, e). Co-treatment of animals with NU9056 prior to cocaine abolished the observed induction of Kat5 in D1R+ MSNs in WT mice (saline/cocaine vs NU9056/cocaine; P = 0.0011), with no effect in ChI-D2RKO (P = 0.7922) (Fig. 7d).

In line with the knowledge that the dorsal striatum is also involved in the addiction process and based on our results showing that NU9056 greatly reduces the psychomotor effects of cocaine (Fig. 7a), we also quantified FISH experiments for Kat5 in the dorsolateral striatum (DLS; Supplementary Fig. S7b, c and Supplementary Dataset S7b, c). In contrast to what was observed for M4R, Kat5 induction following chronic cocaine administration was not restricted to the NAcc but also present in the DLS. However: i) specificity for Kat5 induction in D1R+ MSNs (WT saline/saline vs saline/cocaine; D1R+ MSNs: P = 0.0063, D2R+ MSNs: P = 0.4391); ii) the absence of this induction in ChI-D2RKO animals (ChI-D2RKO saline/saline vs saline/cocaine; D1R+ MSNs: P = 0.9998, D2R+ MSNs: P = 0.5970); and iii) the inhibition of cocaine-induced Kat5 in WT D1R+ MSNs by NU9056 (WT saline/cocaine vs NU9056/cocaine; P = 0.0158) were all retained in the DLS (Supplementary Fig. S7b, c and Supplementary Dataset S7b, c) as observed in the NAcc (Fig. 7d, e). These results further validate the efficacy of Kat5 inhibition by NU9056 in the striatum and support the role of Kat5 induction in cocaine-dependent behavioral and molecular responses.

Finally, to establish a direct connection between Kat5 and H4K8ac, we performed FISH using probes for D1R and D2R coupled to immunofluorescence (IF) with a primary antibody directed against H4K8ac on striatal sections of mice treated contingently for 7 days with cocaine in the presence or absence of NU9056 (Fig. 8). Interestingly, mirroring what was observed for Kat5, a striking cocaine-dependent induction of H4K8ac was observed in D1R+ MSNs (P < 0.0001) (Fig. 8b) of greater intensity than in D2R+ MSNs (P = 0.0201) (Fig. 8c) in the NAcc of WT mice, while again no induction was observed in either MSN subtype in ChI-D2RKO mice (D1R+ MSNs: P = 0.1489, D2R+ MSNs: P = 0.4810) (Fig. 8b, c). Hyperacetylation of H4K8ac in WT D1R+ MSNs was abolished by NU90569 (P = 0.0001) (Fig. 8b), while no significant change in H4K8ac was found in WT D2R+ MSNs in response to co-administration with NU9056 prior to cocaine (P = 0.0833) (Fig. 8c), again highlighting D1R+ MSNs as the primary MSN subtype participating in this mechanism. These results establish a

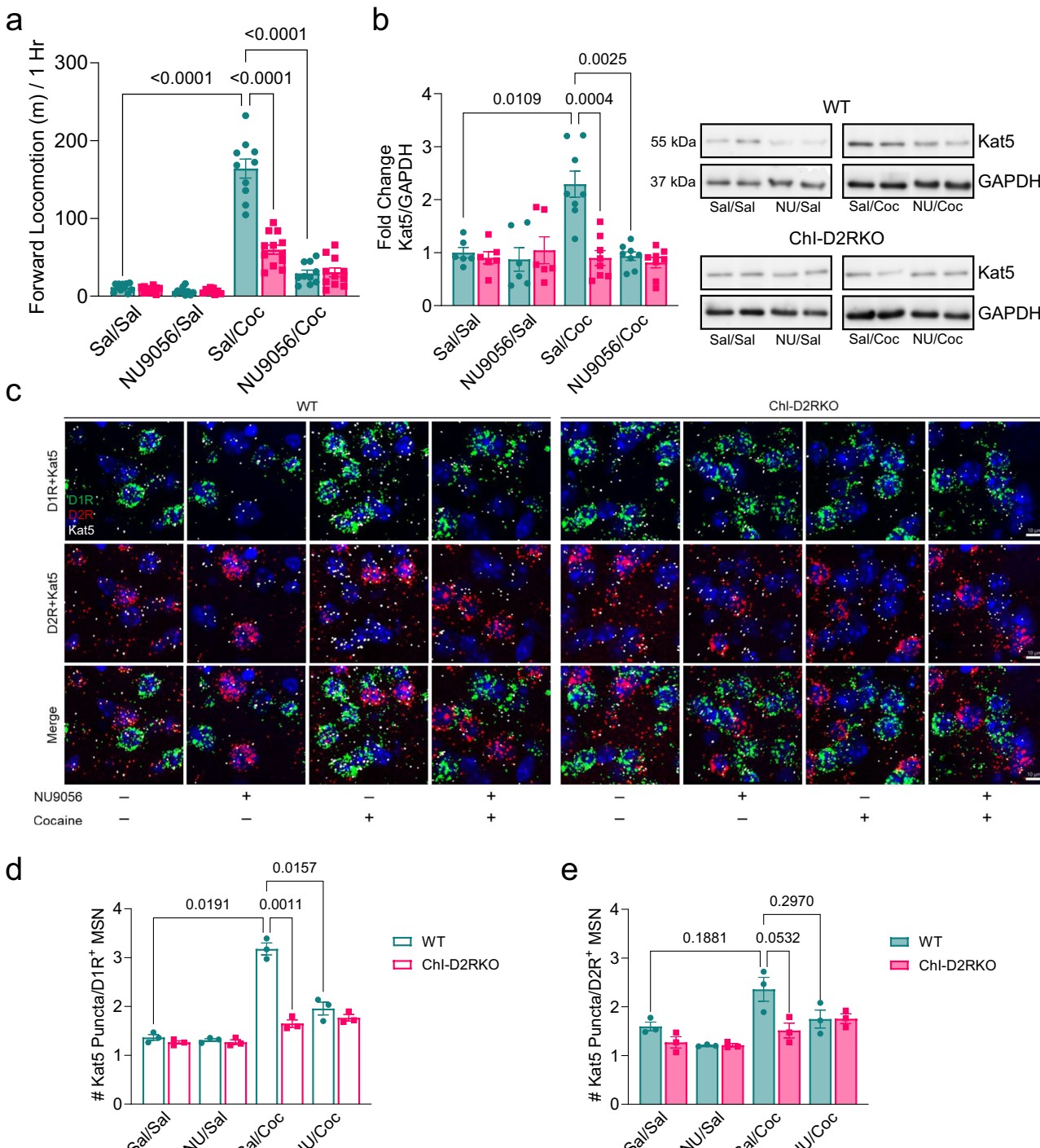

**Fig. 7 | The selective Kat5 inhibitor NU9056 prevents cocaine-induced effects.**
**a** Forward locomotion in WT and ChI-D2RKO mice on the final day of daily chronic treatment with vehicle or NU9056 (2.5 mg/kg i.p.) 1-h prior to saline or cocaine (15 mg/kg i.p). Motor activity was recorded for 1-h following saline or cocaine administration (N = 9–12/group). Two-way ANOVA, genotype: $F_{(1,21)}$ = 43.11, P < 0.0001; treatment: $F_{(1.481,26.65)}$ = 161.8, P < 0.0001; genotype × treatment: $F_{(3,54)}$ = 45.50, P < 0.0001. Tukey post-hoc test, WT sal/sal vs WT sal/coc: P < 0.0001; WT sal/coc vs WT NU9056/coc: P < 0.0001; WT sal/coc vs ChI-D2RKO sal/coc: P < 0.0001. **b** *Left:* Bar graph showing fold change in expression of Kat5/GAPDH by Western blot in NAcc extracts following chronic treatment. Fold changes shown are relative to WT saline (N = 6–8/group). Two-way ANOVA, genotype: $F_{(1,14)}$ = 6.200, P = 0.0260; treatment: $F_{(2.573,29.16)}$ = 10.65, P = 0.0001; genotype × treatment: $F_{(3,34)}$ = 10.96, P < 0.0001. Tukey post-hoc test, WT sal/sal vs WT sal/coc: P = 0.0109; WT sal/coc vs WT NU9056/coc: P = 0.0025; WT sal/coc vs ChI-D2RKO sal/coc: P = 0.0004. *Right:* Representative Western Blot images of samples from *Left*.

**c** Representative images of fluorescent in situ hybridization (FISH) experiments showing D1R (green), D2R (red), and Kat5 (white) expression in the NAcc of WT and ChI-D2RKO mice (N = 3/group) treated with vehicle or NU9056 (2.5 mg/kg i.p.) 1-h prior to cocaine (15 mg/kg i.p.) for 7 days (blue: DAPI). Scale bar: 10 μm.
**d** Quantification of the number of Kat5+ puncta in D1R+ MSNs in the NAcc. Two-way ANOVA, genotype: $F_{(1,4)}$ = 52.85, P = 0.0019; treatment: $F_{(1.988,7.954)}$ = 100.5, P < 0.0001; genotype × treatment: $F_{(3,12)}$ = 44.78, P < 0.0001. Tukey post-hoc test, WT sal/sal vs WT sal/coc: P = 0.0191; WT sal/coc vs WT NU9056/coc: P = 0.0157; WT sal/coc vs ChI-D2RKO sal/coc: P = 0.0011. **e** Quantification of the number of Kat5+ puncta in D2R+ MSNs in the NAcc. Two-way ANOVA, genotype: $F_{(1,4)}$ = 7.627, P = 0.0503; treatment: $F_{(1.809,7.235)}$ = 11.97, P = 0.0056; genotype × treatment: $F_{(3,12)}$ = 4.595, P = 0.0231. Tukey post-hoc test, WT sal/sal vs WT sal/coc: P = 0.1881; WT sal/coc vs WT NU9056/coc: P = 0.2970; WT sal/coc vs ChI-D2RKO data are provided as a Source data file.

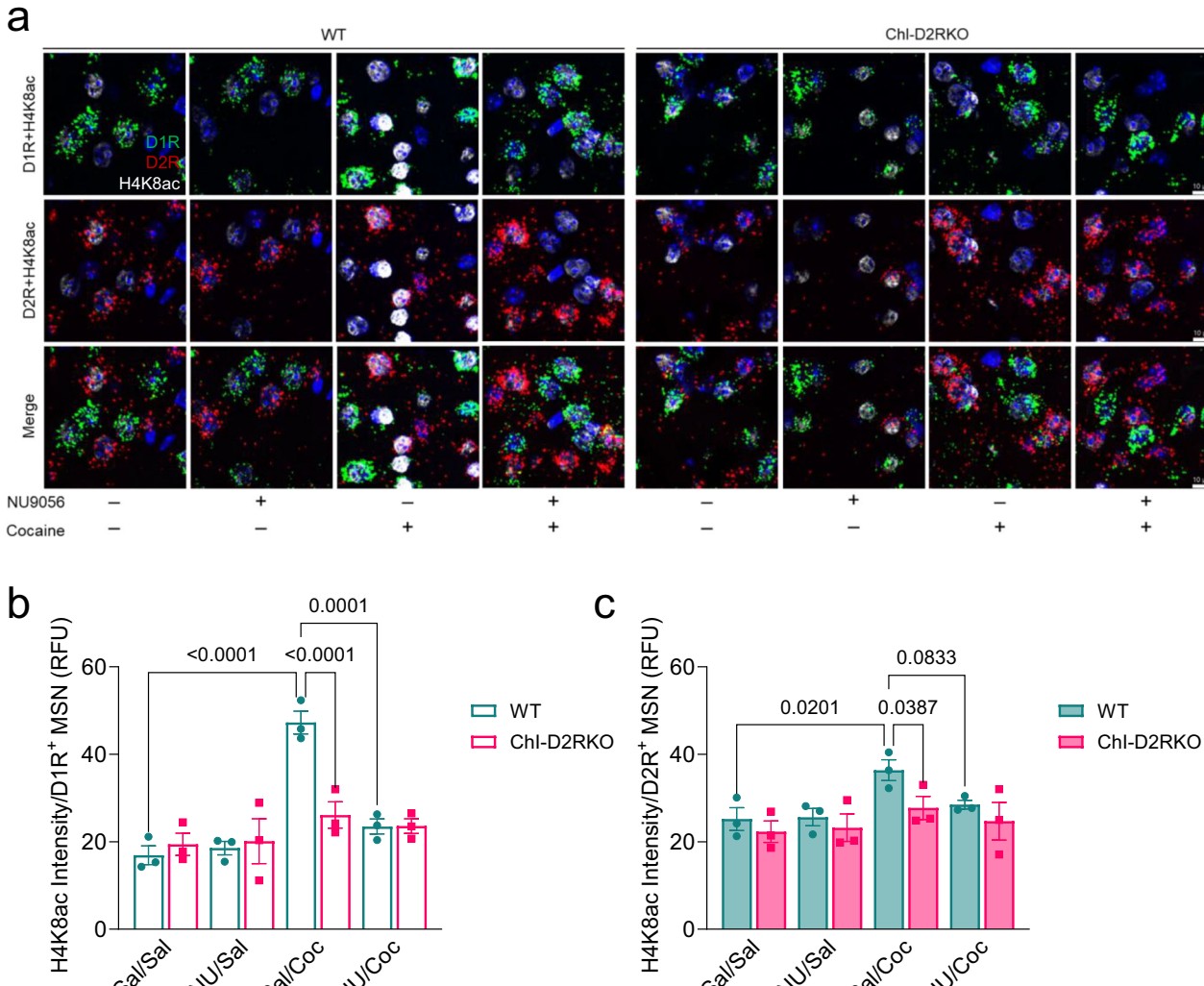

**Fig. 8 | Kat5 inhibition prevents the cocaine-induced acetylation of H4K8.**
**a** Representative images of combination FISH-immunofluorescence experiments showing D1R (green), D2R (red), and H4K8ac (white) expression in the NAcc of WT and ChI-D2RKO mice (N = 3/group) treated with vehicle or NU9056 (2.5 mg/kg i.p.) 1-h prior to cocaine (15 mg/kg i.p.) for 7 days (blue: DAPI). Scale bar: 10 μm.
**b** Quantification of the fluorescent intensity (RFU) of H4K8ac in D1R$^+$ MSNs in the NAcc. Two-way ANOVA, genotype: $F_{(1,4)} = 15.18$, $P = 0.0176$; treatment: $F_{(3,12)} = 15.28$, $P = 0.0002$; genotype × treatment: $F_{(3,12)} = 6.787$, $P = 0.0063$. Tukey post-hoc test,

WT sal/sal vs WT sal/coc: $P < 0.0001$; WT sal/coc vs WT NU9056/coc: $P < 0.0001$; WT sal/coc vs ChI-D2RKO sal/coc: $P < 0.0001$. **c** Quantification of the fluorescent intensity (RFU) of H4K8ac in D2R$^+$ MSNs in the NAcc. Two-way ANOVA, genotype: $F_{(1,4)} = 11.82$, $P = 0.0263$; treatment: $F_{(3,12)} = 3.255$, $P = 0.0597$; genotype × treatment: $F_{(3,12)} = 0.4714$, $P = 0.7079$. Tukey post-hoc test, WT sal/sal vs WT sal/coc: $P = 0.0201$; WT sal/coc vs WT NU9056/coc: $P = 0.0833$; WT sal/coc vs ChI-D2RKO sal/coc: $P = 0.0387$. Values shown are mean ± SEM.

---

direct link between cocaine-induced Kat5 induction and H4K8ac in the striatum of mice.

## Discussion

Dopamine signaling is critical for eliciting the psychomotor and rewarding effects of cocaine, since mice lacking the major DA receptors, D1R or D2R, show altered responses to cocaine[40–43]. DA receptor signaling modulates the activity of striatal circuits, either directly or indirectly, by regulating the release of other neuromodulators/neurotransmitters[18,44–46]. Here, we show that D2R-mediated inhibition of ChI activity is essential for regulating the cholinergic signaling in the striatum, without which cocaine-dependent behavioral and cellular effects are altered. We extend our previous findings showing a blunted response to contingent cocaine in male ChI-D2RKO mice[18] to the subject's motivated response to obtain the drug by IVSA. Importantly, we found that the ability to associate pressing the active lever with the delivery of cocaine infusions in ChI-D2RKO mice does not differ from that of WT mice. Interestingly, however, we observed a significant

decrease in the motivation of ChI-D2RKO mice to work for cocaine infusions with the progressive ratio schedule of reinforcement, which gradually increases the difficulty to obtain the cocaine reward. Thus, the perceived value of cocaine is decreased in male ChI-D2RKO mice as compared to their WT littermates.

This surprising behavioral deficit is linked, at the molecular level, by a large cohort of differentially expressed genes (DEGs) in the NAcc between WT and ChI-D2RKO mice. Among these DEGs identified by our RNA-seq analyses, many are key immediate early transcription factors previously linked to the cellular response to cocaine and addiction, including Fos, Nr4a1, Egr1, and Fosb. Gene ontology analyses comparing sequencing data of the NAcc following cocaine IVSA in WT vs ChI-D2RKO also show that the genes differentially expressed are related to synaptic plasticity, signal transduction, and learning and memory.

Importantly, we identified the HAT, Kat5, as the most significantly differentially expressed histone modifier from our sequencing dataset. The increase of Kat5 after cocaine IVSA likely accounts for the

increased expression of at least some of the upregulated genes in WT animals after cocaine.

Interestingly, our analyses of Kat5 induction under various cocaine regimen at the behavioral, RNA, and protein levels show that this HAT is not only present after IVSA but also following acute or chronic administration of cocaine in WT mice. The consistency of Kat5 induction in both contingent and non-contingent administrations allowed us to perform insightful analyses on the outcomes of increased Kat5 expression after acute and chronic cocaine treatments. To evaluate the behavioral and cellular outcomes of Kat5 induction in striatal responses to cocaine, we used the well-characterized Kat5 inhibitor NU9056. Importantly, we observed that NU9056 administered prior to cocaine prevented the psychomotor response to cocaine in WT mice under both acute and 7-day chronic regimen. These results coincide with the finding that cocaine-induced upregulation of Kat5 is not restricted to the NAcc, but also present in the dorsal striatum, in agreement with the participation of this area in the mechanisms of addiction[47].

Histone acetylation in response to drugs like cocaine controls the expression of genes coding for proteins that participate in drug use and abuse processes[24,25,31]. Analysis of one of the known downstream targets of Kat5, acetylation of histone H4K8, provided further support for the critical role of this epigenetic modification by Kat5 in regulating striatal responses to cocaine exposure. Indeed, ChIP using H4K8ac antibodies showed a robust enrichment of this histone modification at the promoters of Fos, Nr4a1, Egr1, and M4R in WT mice, but not in mice lacking D2R signaling in ChIs. Thus, DA/D2R-mediated inhibitory control of ChIs is critically involved in the epigenetic events that alter the striatal transcriptome at the first encounter with cocaine and later in the addiction process, participating in the motivation for drug use. Future H4K8ac ChIP-seq analyses comparing WT and ChI-D2RKO mice will allow for further, more comprehensive identification of genes under the D2R-mediated dopaminergic control of ChIs.

Our previous studies found that the behavioral and cellular phenotypes of the ChI-D2RKO mice in response to acute cocaine[18] could be restored either by chemogenetic silencing ChIs or through selective blockade of M4R by the muscarinic antagonist tropicamide, suggesting that increased ACh signaling through M4R in ChI-D2RKO mice is responsible for the reduced response to cocaine observed in these mice[18]. In this study, we report that in WT animals, cocaine leads to a specific increase of M4R mRNA expression in the NAcc independent of administration method or duration. This upregulation of M4R expression in response to cocaine is not observed in ChI-D2RKO mice, but administration of tropicamide prior to cocaine successfully restores H4K8ac enrichment on the M4R promoter to WT levels. FISH analyses of M4R induction following contingent chronic treatment with cocaine show that the drug leads to a more potent induction of M4R in the NAcc than in other striatal areas, specifically in D1R+ and not D2R+ MSNs. We believe that this specificity explains the modulatory role of the M4R PAM, VU467154, in cocaine IVSA in WT mice. VU0467154 lowers the volitional cocaine intake during the progressive ratio schedule of reinforcement in the WT animals, which mirrors the phenotype of ChI-D2RKO mice. Indeed, VU0467154 significantly reduces the number of infusions earned and active lever presses in WT mice as compared to vehicle-treated controls. Thus, in the WT, enhancing M4R signaling reduces cocaine intake. That this M4R induction takes place specifically in D1R+ MSNs suggests the presence of a mechanism controlling these neurons response to the drug, where DA/D1R stimulation by cocaine is counterbalanced by ACh/M4R signaling. M4R expression senses the striatal environment and acts as a regulator of ACh signaling in D1R+ MSNs. Interestingly, recent findings have reported M4R downregulation in D1R+ MSNs in the DLS of dopamine lesioned mice[48], which might mimic absence of D2R on ChIs in DLS.

The positive effect of tropicamide in restoring cocaine's effects in ChI-D2RKO mice brought us to suggest a hypercholinergic tone in these mutants. However, direct analyses of ACh levels are needed in support of this hypothesis. Absence of M4R induction by cocaine in ChI-D2RKO mice might be indicative of mechanistic actions to preserve D1R+ MSN functions in the presence of altered ACh levels, and in line with this, M4R expression appears to act as a rheostat to control D1R+ MSN activity. In the absence of inhibitory D2R signaling on ChIs, cocaine might generate a hypercholinergic tone in mutant mice counterbalanced by M4R expression. In agreement, M4R-specific D1R+ MSNs knockout mice show hypersensitivity to stimulants and display more impulsive-like behaviors compared to controls[49,50].

Nevertheless, M4Rs are not only expressed by D1R+ MSNs, acting also as autoreceptors on ChIs as well as heteroreceptors on corticostriatal fibers. Therefore, M4R signaling at any of these locations should be considered in the behavioral and molecular responses to cocaine, both alone and in the presence of pharmacological molecules targeting M4 receptors. Of interest, the presence of M4R on corticostriatal fibers might lead to a reduction of glutamate signaling to striatal neurons in response to cocaine[14,51,52], which might also explain the behavioral and cellular phenotypes observed in ChI-D2RKO mice. The use of the M4R antagonist tropicamide its subsequent effect on corticostriatal terminals could enhance glutamate signaling to striatal neurons, possibly contributing to the observed restoration of normal behavioral and cellular responses to cocaine. Similarly, a higher cholinergic tone can negatively affect nicotinic receptors on dopaminergic neurons, reducing dopamine and possibly glutamate release[53-55]; at the same time, the upregulation of the GABA subunits observed in the mutant's NAcc as compared to the WT (Fig S4) might also suggest aberrant inhibitory signals sent to the VTA in the mutants. A complete evaluation of ACh and DA levels in the NAcc of ChI-D2RKO mice will be a primary focus for future experiments.

In conclusion, D2R-mediated signaling in striatal ChIs plays a key role in regulating the psychomotor effects and motivation for intake of cocaine through the control of ACh release and activation of M4Rs. We speculate that cocaine-dependent dopamine increase to the striatum induces epigenetic modifications driven by Kat5 acetylation of H4K8 and subsequent alteration in the expression of genes that regulate motivation for cocaine intake. The striatal MSN subtype that most strongly shows these modifications is the D1R+ MSNs, where all components of this proposed mechanism converge through M4R-mediated modulation of ACh signaling. With the ability to reduce cocaine intake without impairing motor function, M4R positive allosteric modulators like VU0467154 may have possible clinical utility in the treatment of cocaine use disorders.

## Methods
### Animals
All protocols were submitted and approved by the University of California, Irvine Institutional Animal Care and Use Committee in accordance with the National Institute of Health guidelines. ChI-D2RKO mice were generated by mating D2R$^{floxflox}$ mice[15] (used as WT controls) with choline acetyltransferase (ChAT)-Cre mice, generating D2R$^{floxflox/ChAT Cre/+}$ mice, as previously described[15]. For non-self-administration procedures, animals were group housed and fed *ad libitum*. Adult mice (8–16 weeks in age) were group housed and maintained at standard 12 h/12 h light/dark cycle, at ~25 °C, and humidity levels at 45–60%. For self-administration procedures, adult mice (10–20 weeks in age) were group housed unless fighting was observed, after which mice were separated and single housed for the duration of the experiment. Animals were maintained at reverse 12 h/12 h light/dark cycle, at ~25 °C, humidity levels at 45–60%, and were food restricted to 85–90% of their starting body weight. Self-administration sessions were conducted between zeitgeber times ZT17-ZT22. Male WT and ChI-D2RKO littermates were used for all experiments.

## Drugs

Cocaine HCl (Sigma; Cat# C57761G), Tropicamide HCl (Tocris; Cat# 0909), and NU9056 (Tocris; Cat# 9043) were dissolved in sterile saline (0.9% NaCl, pH 7.4). VU0467154 (MedChemExpress; Cat# HY-112209) was made into a suspension using 10% Tween-80 in sterile $H_2O$ and sonicated at 30% amplitude for 5 s. Intraperitoneal injections were given at a volume of 10 ml/kg.

## Food self-administration

Subjects were mildly food restricted (85–90% of their free-feeding weight) and trained to press a lever in an operant chamber for 20 mg food pellets (5TUM, Test Diet) under a fixed ratio 5, time-out 20 s (FR5TO20s) schedule of reinforcement. Each session was performed using 2 retractable levers (1 active, 1 inactive) for a 1-h session. Completion of the response criteria on the active lever resulted in the delivery of a food pellet and activation of a cue light above the lever for the 20 s time-out duration. Responses on the inactive lever were recorded but had no scheduled consequences.

## Intravenous cocaine and saline self-administration

Subjects were surgically catheterized, as previously described[28,29]. Briefly, mice were anesthetized with an isoflurane (1–3%)/oxygen vapor mixture and prepared with intravenous catheters. Catheters consisted of a 6 cm length of silastic tubing fitted to a guide cannula (Plastics One, Roanoke, VA), bent at a curved right angle, and encased in dental acrylic. The catheter tubing was passed subcutaneously from the animal's back to the right jugular vein, and a 1 cm length of the catheter tip was inserted into the vein and tied with a surgical silk suture. Following the surgical procedure, animals were allowed ≥72 h to recover from surgery, then permitted to acquire intravenous cocaine self-administration during 1 h daily sessions at the standard training dose of cocaine (0.3 mg/kg/infusion)[30], or intravenous saline self-administration as the control condition. Cocaine or saline was delivered through tubing into the intravenous catheter by a Razel syringe pump (Med Associates). Each session was performed using 2 retractable levers (1 active, 1 inactive). Completion of the response criteria on the active lever resulted in the delivery of an intravenous infusion (0.03 ml infusion volume; FR5TO20 sec schedule). Responses on the inactive lever were recorded but had no scheduled consequences. Training duration was determined by the time required to reach a 5-day stabilization of responses to the FR5TO20 schedule for each cohort of mice used (i.e., training was considered complete when the average change in active lever pressing remained under 20% across a 5-day interval). This resulted in different training durations between cohorts, with cohort 1 reaching an average of 18.25% change in a 3-week training period, and cohort 2 reaching an average of 19.53% in a 2-week period. Following the fixed ratio schedule, mice were then transitioned to respond according to the more stringent progressive ratio schedule to assess motivation to obtain the drug[30]. For the progressive ratio, subjects were required to press the active lever with a logarithmic increase in the ratio and the progression of response requirements as follows: 5, 10, 17, 24, 32, 42, 56, 73, 95, 124, 161, 208, 268, 345, 445, 573, 737, 947, 1218, 1566. Maximum time was set to 6 h for the progressive ratio session. Catheters were flushed daily with physiological sterile saline solution (0.9% w/v) containing heparin (100 USP units/ml). Catheter integrity was verified with the ultra-short-acting barbiturate anesthetic Brevital (methohexital sodium, Eli Lilly, Indianapolis, IN) at the end of the study. Behavioral responses were automatically recorded by Med Associates software.

## Novel home cage

Locomotor activity was analyzed and recorded in a novel home cage (NHC) (20 × 30 × 13 cm transparent plastic box) using a video-tracking system (Viewpoint; Lyon, France). Mice were habituated to the NHC for

2 h prior to administration of all treatments. VU0467154 (5 mg/kg i.p.) or vehicle (10% Tween-80 in $H_2O$) and tropicamide (10 mg/kg i.p) or vehicle (saline) were administered 15 min prior to saline or cocaine (15 mg/kg i.p.). NU9056 (2.5 mg/kg i.p.) or vehicle (saline) was administered 1 h prior to saline or cocaine (15 mg/kg i.p.) either in acute or once daily for 7 days (chronic). Motor responses for all treatments were recorded for 1 h following saline or cocaine administration.

## Rotarod

Motor coordination was assessed using a rotarod (Med Associates Inc.; Cat# ENV-577M). Three rotarod training sessions measuring fall latency at 24 RPM were performed 1 h apart from each other. 1 h after the last training session, mice were administered vehicle (10% Tween-80 in $H_2O$) or VU0467154 (5 mg/kg i.p.), and fall latency was measured. All values shown in training and test sessions are the average of 3 trials within sessions.

## Conditioned place preference

The CPP testing apparatus consisted of two compartments (15.5 × 16.5 × 20.3 cm) divided by a neutral space (15.5 × 5 × 20.3 cm); each compartment contained visual and tactile cues on the walls and floors. Each genotype was divided into two groups by conditioning mice to receive either saline or cocaine (10 mg/kg; i.p.) in a specific compartment. On day 1, mice were placed into the apparatus for 20 min and left to explore both sides of the apparatus; activity and time spent in each compartment were recorded and scored. The following day, conditioning started using an unbiased protocol in which the drug-paired compartment was randomly assigned to mice in each group. During conditioning, mice were given either cocaine or saline on alternate days and restricted to the appropriate compartment for 20 min. Cocaine was administered on days 2, 4, and 6 with alternating saline injections on days 3, 5, and 7. On day 8, the CPP test was performed by leaving the mice free to choose between the two compartments for 20 min. Vehicle (10% Tween-80 in H2O) or VU0467154 (5 mg/kg i.p.) was administered 15 min before saline or cocaine treatments on conditioning days 2, 4, and 6.

## Protein preparation and Western blot

Brains were rapidly dissected 1 h after saline or cocaine administration and frozen in 2-methylbutane on dry ice. Hemilateral punches of the NAcc were obtained and sonicated (3 cycles; 30 s ON 30 s OFF; 60% amplitude) in RIPA buffer with a protease inhibitor cocktail (Roche; Cat# 04693132001) and trichostatin A (0.1 μM) as a histone deacetylase inhibitor. Proteins were quantified using a Pierce BCA Protein Assay Kit (Thermo Fisher; Cat# 23225). 5 μg of protein were loaded into a 15% polyacrylamide gel, and electrophoresis was performed. Separated proteins were transferred onto a nitrocellulose membrane, and transfer quality was assessed using Ponceau Red staining. Membranes were washed 3 times with Tris-Buffered Saline + Tween (TBST) (137 mM NaCl, 2.7 mM KCl, 19 mM Tris base, 0.05% Tween-20) for 10 min each and were blocked with 5% BSA in TBST for 1 h. Membranes were then incubated with primary antibodies (Kat5 1:2000, Protein Tech Cat# 10827-1-AP; Alpha Tubulin 1:10000, Protein Tech Cat# 66031-1-Ig; GAPDH 1:10000, Millipore Cat# MAB374) overnight at 4 °C. The next day, membranes were then washed 3 times with TBST for 10 min each and then incubated with secondary antibodies (Anti-rabbit HRP; Cat# AQ132P; Anti-mouse HRP; Cat# AP124P, 1:7500; Millipore). 3 final washes with TBST were done for 10 min each. Immobilon Western Chemiluminescent HRP substrate (Millipore; Cat# P90720) was applied to each membrane for 5 min, and imaging was performed using a Chemidoc (Biorad). Analysis of resulting images was performed using Image Lab Software v 6.0.1 (Biorad). Uncropped and unprocessed scans are provided in the Source data File.

## In situ mRNA expression

WT and ChI-D2RKO mice (N = 3/group) were treated daily for 7 days with vehicle (saline) or NU9056 (2.5 mg/kg i.p.) 1 h before saline or cocaine (15 mg/kg i.p.). 1 h after the final saline or cocaine administration, mice were deeply anesthetized with Euthasol (pentobarbital sodium and phenytoin sodium; Virbac) prior to transcardial perfusion with 10 mL 1X PBS followed by 30 mL 4% paraformaldehyde in PBS. Whole brains were extracted, post-fixed overnight in 4% paraformaldehyde in PBS, cryoprotected in stepwise sucrose incubations at 10%, 20%, and 30% sucrose in PBS, then embedded in OCT (Sakura) and frozen at −80 °C for cryostat (Leica CM1950) sectioning. 10 µm coronal striatum sections were collected serially and mounted directly onto Superfrost Plus slides (Fisher Scientific) such that each slide stained contained 4 sections equally spaced and spanning the region from approximately +1.5 mm to +0.5 mm relative to bregma, allowing for a more composite analysis of the striatum. In situ mRNA expression was performed using the RNAscope Multiplex Fluorescent Reagent Kit v2 (Advanced Cell Diagnostics Cat# 323270) according to manufacturer instructions for fixed-frozen tissue. In situ in combination with protein immunofluorescence was performed using the RNAscope Multiplex Fluorescent Reagent Kit v2 in combination with the RNA-Protein Co-detection Ancillary Kit (Advanced Cell Diagnostics Cat# 323180) according to manufacturer instructions for fixed-frozen tissue, along with primary antibody for H4K8ac (1:300, Active Motif Cat# 61103) and AlexaFluor 647 anti-rabbit secondary antibody (1:750, Invitrogen Cat# A-21245). RNAscope probes and TSA Vivid Dyes were used as follows: Drd1-C3, TSA Vivid 520 1:2000; Drd2-E2-C2, TSA Vivid 570 1:1000; Chrm4-C1, TSA Vivid 650 1:2000; Kat5-O1-C1, TSA Vivid 650 1:1500. Slides were coverslipped with ProLong Gold Antifade Mountant (Invitrogen Cat# P36934) and allowed to cure overnight at room temperature before storage at 4 °C and imaging.

## Image acquisition and analysis

63× dorsal and ventral striatum images of fluorescent in situ hybridization and RNA-protein co-detection experiments were acquired using a confocal microscope (Leica SP8, UC Irvine Optical Biology Core), and all analyses were performed using Imaris 10 software (UC Irvine Optical Biology Core). For analysis of D1R, D2R, M4R, and Kat5 puncta, a Surface mask of DAPI was created to identify individual cells, and DAPI masks corresponding to D1R$^+$ and D2R$^+$ MSNs were manually selected for and individually duplicated to generate separate channels for each cell type to be analyzed. The Cells function was then used to quantify puncta counts and fluorescent intensity in each cell as follows: the cell body detection source channel was selected as either the D1R or D2R corresponding DAPI mask, cell absolute intensity threshold was set to 10, touching nuclei were attempted to be split using a seed point diameter of 4 µm, cells were selected for using a quality threshold of 8 and minimum voxel size of 1000, and finally puncta were detected as vesicles according to the appropriate source channel and filtered each with an estimated diameter of 1 µm and region threshold of 75. For analysis of H4K8ac fluorescent intensity, a Surface mask of H4K8ac was created, and masks corresponding to D1R$^+$ and D2R$^+$ MSNs were again manually selected for and individually duplicated to generate separate channels. To account for overlapping cells and cells with incomplete nuclear coverage of H4K8ac staining, a morphological split with a 1.5 µm seed point diameter was applied during Surface creation, and split surfaces within the same cell were unified manually. Cells with overlapping nuclei that were not properly split and cells not fully in frame were manually deleted and excluded from analysis. For all quantifications, an average was calculated across 4 sections analyzed for each mouse (N = 3/group).

## Chromatin Immunoprecipitation

Unilateral punches of the ventral striatum from frozen brains of two mice were pooled per sample (3–4 mice/group). Tissue homogenized using an electric homogenizer for 20 s. The samples were then double crosslinked with disuccinimidyl glutarate (2.0 mM) for 30 min and formaldehyde (1%) for 15 min followed by adding glycine (0.125 M) at room temperature for 10 min. Samples were centrifuged at 1200 RPM for 10 min, and pellets were resuspended in 1 mL of hypotonic lysis buffer (10 mM HEPES, pH 9.0, 85 mM KCl, 0.5% IGEPAL), and slowly mixed on an inverter for 10 min at 4 °C. Nuclei were centrifuged at 1200 RPM at 4 °C, and pellets were resuspended in sonication/IP buffer (1X PBS, 1% IGEPAL, 1% NaDOC, 0.2% SDS). Sonication was done using a Bioruptor Pico (10 cycles, 30 s ON/30 s OFF) to generate 200–500 base pair fragments. Supernatants were diluted in the sonication/IP buffer. 20 µL of Protein G Magnetic beads (Cell Signaling; Cat# 70024) were incubated with 2.5 µg of anti-H4K8ac antibody (Active Motif; Cat# 61103) overnight at 4 °C. Diluted samples were then incubated with the antibody bead complex overnight at 4 °C. To monitor the specificity of ChIP assays, samples were also immunoprecipitated with Rabbit isotype-matched control immunoglobulin (Invitrogen; Cat# 10500C). Beads were recovered, washed in low salt buffer (100 mM Tris-HCl pH 7.5, 250 mM LiCl$_2$, 1% IGEPAL, 1% NaDOC) twice and once with high salt buffer (100 mM Tris-HCl pH 7.5, 500 mM LiCl$_2$ 1% IGEPAL, 1% NaDOC). Elution buffer (300 mM NaCl, 0.5% SDS, 10 mM Tris-HCl, 5 mM EDTA) was added to the washed beads, treated with RNase at 37 °C for 2 h and Proteinase K at 65 °C overnight. DNA was recovered using a ChIP DNA kit following the manufacturer's protocol (Active Motif; Cat# 58002). Quantitative PCRs were performed using PowerUP SYBR Green Supermix (Biorad; Cat# 172-5270) according to the manufacturer's protocol. Samples were set as a percentage of input. Primers used for ChIP analysis by RT-PCR are the following:

Fos Promoter Forward: 5′-TACGACCCTTCAGGCATAC-3′,
Fos Promoter Reverse: 5′-GTTTTAAAGGACGGCAGCAC-3′;
Nr4a1 Promoter Forward: 5′-ATTTACAACACCCCTCCTCC-3′,
Nr4a1 Promoter Reverse: 5′-TTCCATTGACGCAGGAGCG-3′;
Egr1 Promoter Forward: 5′-CGTCACTCCGGGTCCTCCCG-3′,
Egr1 Promoter Reverse: 5′-AGGGTCTGGAACAGCAGGGGCC-3′;
M4R Promoter Forward: 5′-GCCGTGGAGAGGTTCGAAAA-3′;
M4R Promoter Reverse: 5′-CAGTCGCCAGGGAGGATTAG-3′;
Gapdh Promoter Forward: 5′-ACCCTGTGTCCACGAGGGCA-3′,
Gapdh Promoter Reverse: 5′-CCTAAGCTGGGACCCCGGGG-3′

## Library preparation for RNA-sequencing

Brains were rapidly dissected 1 h after the last session and frozen in 2-methylbutane on dry ice. Bilateral tissue punches of the ventral striatum were obtained and rapidly homogenized in TRIzol (Invitrogen; Cat# 15596026). Total RNA was monitored for quality control using the Agilent Bioanalyzer Nano RNA chip and Nanodrop absorbance ratios for 260/280 nm and 260/230 nm. Library construction was performed according to the Illumina TruSeq® Stranded mRNA Sample Preparation Guide. The input quantity for total RNA was 500 ng, and mRNA was enriched using oligo dT magnetic beads. The enriched mRNA was chemically fragmented for 3 min. First-strand synthesis used random primers and reverse transcriptase to make cDNA. After second-strand synthesis, the ds cDNA was cleaned using AMPure XP beads, and the cDNA was end-repaired, and then the 3′ ends were adenylated. Illumina barcoded adapters were ligated on the ends, and the adapter-ligated fragments were enriched by nine cycles of PCR. The resulting libraries were validated by qPCR and sized by Agilent Bioanalyzer DNA high-sensitivity chip. The concentrations for the libraries were normalized and then multiplexed together. The multiplexed libraries were sequenced on paired-end 100 cycles chemistry on the Novaseq 6000. The version of Novaseq control software was NVCS ver 1.7.0 with real-time analysis software, RTA 3.4.4.

## Bioinformatics

Read quality was assessed by FastQC (www.bioinformatics.babraham.ac.uk/projects/fastqc/).

Sequence alignment was performed using the *mus musculus* GENCODE reference genome (GTF file from release M23 GRCm38.p6) using STAR 2.6.0c software[56]. Annotation and count matrices were generated using Genomic Features and Genomic Alignments[57]. Differential expression analysis was performed using DESeq2[58]. Read counts were normalized using the relative log expression method of DESeq2. Normalized read counts were converted into the log-read counts, which were then used for identifying differentially expressed genes; in our comparisons saline-treated WT mice were used as the control. Unadjusted p-values were determined for each gene; $p < 0.05$ was considered statistically significant. Volcano plots were generated using the ggplot R package. Biological processes represented by the differentially expressed genes were determined using PANTHER Gene Ontology. Genes with $p < 0.05$ and $Log_2$ Fold Change greater than 0.3 and less than −0.3 were entered into PANTHER Gene Ontology; Benjamini test $p < 0.05$ was considered significant. RNA-seq data are publicly available on ArrayExpress (E-MTAB-14303).

## Statistical analyses

Statistical analyses were performed using GraphPad Prism software (version 10.2.3, GraphPad Software, San Diego, CA, USA). The probability of a type I error (significance level) was set at $\alpha = 0.05$ and of a type II error set at $\beta = 0.10$, to give a power (confidence level; 1-β) of 0.90. One-way, two-way, and three-way analysis of variance (ANOVA) was conducted to assess the differences among multiple groups. Tukey's multiple comparison test was used for post-hoc analysis to determine specific group differences when the ANOVA yielded a significant result. Student's t-test, either one-tailed or two-tailed as appropriate, was utilized for pairwise comparisons between two groups. The choice between one-tailed and two-tailed tests was determined based on the specific hypotheses being tested and the directionality of the expected effects. Measurements and quantifications of each experiment were taken from distinct samples except in cases of chronic or IVSA behavioral experiments in which recordings for the same mice were repeated over multiple days.

## Reporting summary

Further information on research design is available in the Nature Portfolio Reporting Summary linked to this article.

# Data availability

Raw data for RNA-seq analysis are available on the NCBI Sequence Read Archive (SRA) under accession number PRJNA1338717. Source data's files are provided with this paper. Source data are provided with this paper.

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

## Acknowledgements

The authors thank all the members of the laboratories for scientific discussion and technical assistance. The authors greatly appreciate Melanie Oakes and Seung-Ah Chung at the UCI Genomics High-Throughput Facility for their RNA-sequencing support. This study was made possible in part through access to the Optical Biology Core Facility of the Developmental Biology Center, a shared resource supported by the Cancer Center Support Grant (CA-62203) and Center for Complex Biological Systems Support Grant (GM-076516) at the University of California, Irvine. R.G.L. was supported by a University of California, Irvine School of Medicine Dean's Fellowship, the Dr. Lorna Carlin Scholar Award, and a University of California, Irvine Graduate Dean's Dissertation Fellowship; R.G.L. and L.O. were recipients of a T32 Predoctoral Training Grant (DA050558) from the NIH NIDA. Y.S. was supported by NIH NIDA F31 Training Grant (DA050436), and V.L. by NIH NIDA grant (DA039658 to C.D.F.). This work was supported by grants from NIH (DA035600 to E.B.) and by the French Institut National de la Santé et de la Recherche Médicale (INSERM).

## Author contributions

R.G.L. and E.B. designed the study. R.G.L., L.O., D.P., E.F., and T.D.N.P. performed behavioral and cellular studies; R.G.L., Y.S., and V.L. performed, and C.D.F. supervised self-administration studies. R.G.L. performed RNA-seq analyses, and M.Z. and M.S. performed data analytics. R.G.L., T.D.N.P., and E.F. performed ChIP-qPCRs and qPCRs. L.O. performed and analyzed RNAscope experiments. R.G.L. and L.O. analyzed the data, and E.B. supervised the study. E.B. wrote the manuscript with R.G.L. and L.O.

## Competing interests

C.D.F. serves on the scientific advisory board of a therapeutic development company, GATC Health, which is unrelated to the research presented herein. All the other authors declare no competing interests.
