## [Transparent Peer Review file · Nature Communications]

Epigenetic regulation of cocaine intake through dopaminergic control of cholinergic interneurons in male mice

Corresponding Author: Professor Emiliana Borrelli

Version 0:

Reviewer comments:

Reviewer #1

(Remarks to the Author)

This is a very nice study in which the authors describe an important role for dopamine-regulation cholinergic interneurons (via D2 receptors) on downstream behavioral and transcriptomic responses to cocaine. They specifically implicate KAT5, a histone acetyltransferase, in nucleus accumbens – downstream of M4 receptor signaling – in mediating these effects. The study nicely integrates behavioral, pharmacological, and molecular approaches. It is well written, the figures are clear, and the data are compelling. I offer the following suggestions which would further strengthen an already strong study.

The authors use published datasets to implicate dMSNs in the elaborated mechanisms. It would be very helpful if the authors could provide direct experimental evidence for this interpretation. For example, they could use RNAscope for *Drd1* or *Drd2* (good albeit imperfect markers for dMSNs or iMSNs) combined with immunohistochemistry or RNAscope for KAT5/Kat5 as well as immunohistochemistry for H4K8ac (which should work well with existing antibodies) to demonstrate cocaine induction of KAT5 and H4K8ac in *Drd1*+ cells but not in other nucleus accumbens cell types.

The authors use ChIP to nicely demonstrate H4K8ac on key immediate early genes implicated in cocaine action. It would be powerful to complement these findings with a global analysis of H4K8ac by use of ChIP-seq or CUT&RUN. On the other hand, this is a large undertaking and it is acceptable if the authors would prefer to simply mention this as a goal for future studies.

Reviewer #2

(Remarks to the Author)

The authors have conducted investigations mainly using Chl-D2RKO mice, which would be a building on their previous reports (Kharkwa et al, 2016; Lewis et al, 2020). They previously reported that *Drd2*-mediated Chl inhibition underlies cocaine-induced response, and now showed that through IVSA cocaine-mediated motivation would be compromised in Chl-D2R KO mice compared with WT mouse. Importantly, they argued that *Kat5*-induced H4K8 acetylation would elicit cocaine responses including augmented motivation, which is impaired in Chl-D2R KO mice. The epigenetic insights for addictive states defined by drug self-administration would be somewhat novel. However, various important data are missing, which cannot fully support their arguments. Given a number of previous reports indicating epigenetic regulation of addiction behaviors, this manuscript did not reach novelty that merits publication of this journal.

The authors performed bulk RNA-seq, which inherently has only limited capability to analyze specific cell types. To complement this caveat, they calculated correlations and inferred that the *Kat5*-related mechanisms occur prominently in dMSNs. This argument, however, did not seem to be strong enough, falling short of the clear validity. In addition, the methodology for calculating these correlations and establishing their significance was not clearly explained. Besides Figure 2D, no additional evidence was provided to confirm that *Kat5*-mediated regulation took place mainly in the given cell type, dMSNs. It would be beneficial if they attempted further imaging tools such as immunohistochemistry or RNAscope.

The authors claim that the abnormal behaviors that Chl-D2RKO mice exhibited are due to *Kat5* downregulation in the NAc,

but it remains unclear that they are attributable exclusively to changes in NAc circuits. They would compare other striatal regions such as the dorsal striatum.

The authors argue that M4R overactivation in Ch1-D2RKO mice can prevent cocaine-induced Kat5 expression. But this reviewer found neither direct & convincing evidence for activation and induction of M4R except a simple histogram, and nor tangible evidence indicating that M4R could be activated in absence of Drd2 (molecularly and cellularly). They should provide experimental evidence for this issue as it seems to be one of major points throughout this whole story.

The authors compared WT/Ch1-D2R KO groups after cocaine IVSA and then performed RNA-seq, and they also stated Kat5/M4R would be crucial in the development of motivated responses for cocaine, but not positive reinforcement. However, in the following data, they mostly performed acute-cocaine administration and compared the data to observe the role of Kat5/M4R in cocaine-mediated responses. They should carry out IVSA to explore the attributes of M4R/Kat5 to cocaine motivation clearly. These should be resolved.

Target mechanisms were assayed and validated mainly by perfusion of various drugs (NU9056, tropicamide, and VU04567154), all of which were administered via intraperitoneal (IP) injection. However, it would be questionable whether the same effects could be observed if these drugs were administered in a brain-region-specific or cell-type-specific manner. Additionally, except VU04567154, the effects of those drugs were only tested through locomotion and cocaine-conditioned place preference (CPP). Since cocaine self-administration is a most direct method for assessing cocaine motivation as they reckoned, they should examine whether these drugs have any impact on motivation using IVSA.

There are a number of minor points as follows:

Fig 1. The authors have not confirmed whether the knockout effect of Ch1-DRD2KO mouse was confined to the NAc. Experiments for region-specific deletion of DRD2 in Ch1, or rescuing overexpression of DRD2 in the NAc of Ch1- D2RKO mice would be required.

Fig 2. When observing transcriptomic changes after PR, the number of cocaine infusions was already different. This would necessitate additional experiment with control group that received the same amount of cocaine using a non-contingent paradigm.

Fig 4. Since Kat5 would be expected to work across whole brain, it should be depleted (shown in ref 30, See Allen brain ISH data #69735174) or at least drugs should locally delivered via intracerebral infusion to target brain areas.

Can the effect of the Kat5 inhibitor be conclusively determined from capturing Kat5 in western blot? Capturing histone acetylation (substrate of Kat5) via western blot or IHC would be more straightforward (shown in ref 30). Typo on line 216, Fig 4B should be 3B

Fig 5. Typo on line 264, Fig 4C should Fig 5C

Reviewer #3

(Remarks to the Author)

The study by Lewis et al. identified deficits in cocaine seeking and cocaine-induced long-lasting molecular changes in the NAc in mice lacking D2R in Ch1s. They unveiled that the absence of D2R in Ch1s prevents the upregulation of histone acetyltransferase, KAT5 in the NAc of cocaine-treated mice. This event is accompanied by a reduction of the acetylation of histone H4 on lysine 8 normally detected in the promoters of various IEGs. They finally designed a series of experiments aiming to demonstrate that these effects are due to a hypercholinergic state in the Ch1-D2RKO. Overall, the study is well-designed and adds to an important topic in the field. Despite my excitement about the novel insights explored in this study, several issues interfere with the correct interpretation of the findings. Below is a list of additional experiments to be performed by the authors to strengthen the conclusion of the present work.

Major concerns

Intro: Second paragraph : the notion of direct and indirect pathway refers to the dorsal striatum not to the NAc. Accumbal D1R-expressing MSNs do not project to the SN but to the VP and to lesser extent to the VTA. Please correct and quote appropriate references.

Fig 1: It is not clear whether male and/or female mice are used. SABV is neither considered nor is the combination of male and female mice into a single data set justified if applicable. This aspect is crucial considering that behavioral and molecular alterations mediated by cocaine strongly differ between male and female mice (PMID: 34272455 ; PMID: 28072417). This issue should be clearly addressed. In Fig S2B, both wt and Ch1-D2RKO press more on the active lever delivering saline. Does it mean that saline self-administration is reinforcing in mice? Please clarify this issue.

Fig 2: Panel B. It is not clear from the text and the volcano plot what these DEGs represent: a difference between saline vs cocaine independently of genotypes or among all genes downregulated by cocaine, 897 genes are different between wt and Ch1-D2RKO. Please clarify the text and include additional RNAseq analyses. Related to my previous comments. If both male and female mice are used, how the authors manage to integrate potential sex-differences in their analyses? The analysis in panel D is original but cell-type specific increased expression of Kat5 must be confirmed by smFISH with probes

allowing the identification of MSNs, ChIs and other classes of GABAergic interneurons. This should be compared to ChI-D2RKO. This control is crucial if one wants to understand at which circuit level the lack of D2R in ChI impact the expression of Kat5.

Fig 3: The increase of KAT5 expression (at protein level) detected only one hour after a single cocaine administration is impressive. Immunofluorescence analysis of KAT5 expression in D1tomato/EGFP and D2eGFP mice should be performed to complete the characterization. This level of analysis would also precise whether increased KAT5 expression occurs in NAc Core and/or Shell. If the antibody does not allowed such level of analysis, this could be solved by smFISH (see comment Fig 2). Please indicate in the text whether ChIP experiments were performed on NAc extracts from mice acutely administered with cocaine or after IVSA.

Fig 4: Panel B. It is not clear from the text what these WB represent. If color codes correspond to the ones on panel A then these results deserved to be explained. If I understand well, the increased expression of KAT5 by cocaine (acute or chronic?) depend on the activity of KAT5? If so, the authors should at least demonstrate a H4K8Ac enrichment at the promoter of Kat5 following cocaine administration. In the same line, the authors should demonstrate that the H4K8Ac enrichment at the promoter of Fos, Nr4a1 and Egr1 is causally linked to the increased of KAT5 expression using NU9056.

Fig 5: Please confirm the increased of Chrm4 by smFISH to determine in which striatal-cell type this regulation occurs. This result will be important for the interpretation of the data obtained with the M4R antagonist. Please precise whether these data have been generated on NAc extracts from mice acutely administered with cocaine. I am not sure that based on these results we can conclude we have a hypercholinergic phenotype. In their reasoning, the authors seem to have totally neglected two important points. The first one refer to the heterogeneity of ChIs responses due to regional variation in dopamine-neuron glutamate cotransmission "Dopamine neurons drive a burst-pause firing sequence in cholinergic interneurons in the medial shell of the nucleus accumbens, mixed actions in the accumbens core, and a pause in the dorsal striatum." (PMID: 24559678). The second important aspect is related to the fact that ChIs express VGLUT3 and use both acetylcholine and glutamate as neurotransmitters, a property crucial for behavioral and cellular effects mediated by cocaine (PMID: 26239290). Knowing the importance of this co-transmission, the impact of the absence of D2R signaling in ChIs on the Ach/Glu co-transmission should be addressed and integrated in the interpretation of results.

Fig 6 & 7: CPP measures the rewarding property of a drug not the reinforcing property (p12 line 276 / p13 line283 please correct). Reinforcement refer to instrumental learning. IVSA allows the evaluation of the reinforcing property of cocaine. In Fig 7 panel C, the breaking point in control mice is much lower than the ne reported in Fig 1. Could the authors comment on this? The authors should be more cautious with their conclusion. They cannot infer what happens in ChI-D2RKO based on what they observed in control mice treated with M4R PAM and without addressing the co-transmission issue (see above).

Discussion should be expanded to address raised inconsistencies in the Results. However, it is advisable to rewrite the discussion regarding the behavioral experiments, pending suggested experiments and analyses.

Minor concerns

- p3 line 60 : Please quote papers related to mouse nucleus accumbens. The present refs are not really appropriate.
- p8 line 174: Figure 2A should be called not 2C. Please correct.
- p10 line 216: Figure miscalled. Fig 3B should be called not 4B. Same changes should be done for the paragraph.
- p12 line256: Figure miscalled. Fig 5B should be called not 4B. Same changes should be done for the paragraph.

Version 1:

Reviewer comments:

Reviewer #1

(Remarks to the Author)

I was very enthusiastic about the original version of this study and find the revised version that much stronger. I thank the authors for attending to cell-type-specific mechanisms of Kat5. The manuscript is acceptable for publication in its present form and will be a strong contribution to the literature.

Reviewer #2

(Remarks to the Author)

Thank you for your thoughtful revisions. The manuscript has improved, but some key concerns regarding mechanistic specificity and causality in the proposed epigenetic-behavior model remain unresolved. Below are the major points requiring attention:

1. While the RNAscope data (Figs. 5, 7) elegantly demonstrate correlative upregulation of Kat5 and M4R in D1R+ MSNs, the claim of cell-type-specific necessity lacks functional validation. To strengthen causality, this reviewer suggests:

Cell-type-specific deletion/depletion of Kat5 (e.g., using D1-Cre or D1-FLP mice).

Rescue experiments (e.g., M4R overexpression in D1-MSNs to test sufficiency).

These experiments would directly test whether Kat5/M4R functions in D1-MSNs would be selective and necessary for the observed phenotypes.

2. The model hinges on the premise that D2R deletion in cholinergic interneurons (ChINs) elevates tonic ACh release, but direct evidence is missing. This reviewer suggests:

GRAB_ACh sensor data to quantify dynamic ACh changes in the striatum.

HPLC-based ACh measurements in ChIN-D2R KO mice under relevant conditions.

Without this, the link between D2R loss, ACh release, and M4R activation remains speculative.

3. The behavioral effects of M4R/Kat5 manipulation are compelling but do not yet prove that these molecular changes are necessary or sufficient in imparting the observed phenotypes to the ChIN-D2R KO mice. Moreover, the epigenetic and behavioral consequences seem consistent with their observation but remain speculative. Addressing these questions with additional experimental data would solidify the proposed mechanistic chain.

While the revision represents progress, the manuscript would significantly benefit from these additional experiments to resolve lingering mechanistic uncertainties. Targeted functional studies (as outlined above) would transform correlative observations into a more causal model, aligning with standards of this journal for mechanistic insights.

REVIEWER COMMENTS

Reviewer #1 (Remarks to the Author):

This is a very nice study in which the authors describe an important role for dopamine-regulation cholinergic interneurons (via D2 receptors) on downstream behavioral and transcriptomic responses to cocaine. They specifically implicate KAT5, a histone acetyltransferase, in nucleus accumbens – downstream of M4 receptor signaling – in mediating these effects. The study nicely integrates behavioral, pharmacological, and molecular approaches. It is well written, the figures are clear, and the data are compelling. I offer the following suggestions which would further strengthen an already strong study.

- We thank this reviewer for the positive comments on our study. We have well taken criticisms and suggestions. We addressed the criticisms by performing additional experiments, as detailed in the following text.

The authors use published datasets to implicate dMSNs in the elaborated mechanisms. It would be very helpful if the authors could provide direct experimental evidence for this interpretation. For example, they could use RNAscope for Drd1 or Drd2 (good albeit imperfect markers for dMSNs or iMSNs) combined with immunohistochemistry or RNAscope for KAT5/Kat5 as well as immunohistochemistry for H4K8ac (which should work well with existing antibodies) to demonstrate cocaine induction of KAT5 and H4K8ac in Drd1+ cells but not in other nucleus accumbens cell types. The authors use ChIP to nicely demonstrate H4K8ac on key immediate early genes implicated in cocaine action. It would be powerful to complement these findings with a global analysis of H4K8ac by use of ChIP-seq or CUT&RUN. On the other hand, this is a large undertaking and it is acceptable if the authors would prefer to simply mention this as a goal for future studies.

- Following these suggestions, we have removed the figure in which we compared published dataset with our RNA-seq, and performed FISH using RNAscope to assess the colocalization of D1R and D2R with KAT5 and M4R. We found that KAT5 is expressed in both MSNs types (Fig. 7). However, cocaine significantly induces Kat5 only in D1R⁺MSNs (Fig. 7), while only a trend to an increase but of lower intensity is observed in D2R⁺MSNs. Importantly, cocaine induces KAT5 in the WT striatum, but not in the ChI-D2RKO (Fig. 7). We also performed M4R analyses using RNAscope and found that M4R expression was induced by cocaine in the WT, but not in the ChI-D2RKO, in support of our RNA-seq analyses. As requested, we also performed immunofluorescence coupled to FISH (RNAscope), using antibodies directed against H4K8ac. We observed that H4K8ac's IF mirrors M4R RNA induction and is restricted to D1R⁺MSNs of WT, and absent in ChI-D2RKO mice (Fig. 8). Thus, we thank this reviewer for suggesting these experiments that strengthen our findings. Furthermore, we also performed ChIP using H4K8ac antibodies on striatal extracts for M4R and found

its enrichment on the M4R promoter. We added a sentence on ChIP-seq as future objectives in the discussion.

Reviewer #2 (Remarks to the Author):

The authors have conducted investigations mainly using Chl-D2RKO mice, which would be a building on their previous reports (Kharkwa et al, 2016; Lewis et al, 2020). They previously reported that Drd2-mediated Chl inhibition underlies cocaine-induced response, and now showed that through IVSA cocaine-mediated motivation would be compromised in Chl-D2R KO mice compared with WT mouse. Importantly, they argued that Kat5-induced H4K8 acetylation would elicit cocaine responses including augmented motivation, which is impaired in Chl-D2R KO mice. The epigenetic insights for addictive states defined by drug self-administration would be somewhat novel. However, various important data are missing, which cannot fully support their arguments. Given a number of previous reports indicating epigenetic regulation of addiction behaviors, this manuscript did not reach novelty that merits publication of this journal.

The authors performed bulk RNA-seq, which inherently has only limited capability to analyze specific cell types. To complement this caveat, they calculated correlations and inferred that the Kat5-related mechanisms occur prominently in dMSNs. This argument, however, did not seem to be strong enough, falling short of the clear validity. In addition, the methodology for calculating these correlations and establishing their significance was not clearly explained. Besides Figure 2D, no additional evidence was provided to confirm that Kat5-mediated regulation took place mainly in the given cell type, dMSNs. It would be beneficial if they attempted further imaging tools such as immunohistochemistry or RNAscope.

- This criticism was well taken, and we did perform RNAscope as suggested (please see answer to Rev.# 1 and Fig. 7)

The authors claim that the abnormal behaviors that Chl-D2RKO mice exhibited are due to Kat5 downregulation in the NAc, but it remains unclear that they are attributable exclusively to changes in NAc circuits. They would compare other striatal regions such as the dorsal striatum.

- In the article we focused on the NAcc because the RNA-seq was performed on tissue punches from this area, which is the region mostly involved in the rewarding properties of cocaine. However, we recognize that also the dorsal striatum is involved in the addiction process and thus in the present version we have quantified gene expression in both areas. The results are shown in Fig. S7b-c, Kat5 is also significantly induced in D1R+MSNs by cocaine of the WT DLS, but not in that of the Chl-D2RKO. Therefore, Kat5 induction is not restricted to the NAcc but also present in the DLS. This agrees with the effects of the Kat5 inhibitor's NU9056, which prevents Kat5 activity and blocks the motor response to cocaine (Fig. 7).

The authors argue that M4R overactivation in Chl-D2RKO mice can prevent cocaine-induced Kat5 expression. But this reviewer found neither direct & convincing evidence for activation and induction of M4R except a simple histogram, and nor tangible evidence indicating that M4R could be activated in absence of Drd2 (molecularly and cellularly). They should provide experimental evidence for this issue as it seems to be one of major points throughout this whole story.

- We have well taken this point and analyzed M4R expression by RNAscope in WT and Chl-D2RKO. These data are shown and quantified in Fig. 5a-e, in which a significant increase of M4R expression in the D1R⁺ MSNs, upon cocaine treatment, is observed in the WT, but not in Chl-D2RKO. In addition, to address the previous comment, we also analyzed M4R expression in the dorsal striatum and found that M4R activation in response to cocaine is restricted to the NAcc (Fig. S6). At the molecular level, we show that the reported induction of immediate early genes by cocaine is present in the WT but not in the Chl-D2RKO. Induction of IEGs in D1R⁺MSNs is dependent on the activation of these cells by dopamine through the cAMP/PKA pathway. We show that these genes are not upregulated in the Chl-D2RKO as compared to WT. M4R activation contrasts D1R signaling by inhibiting the cAMP/PKA pathway. IEGs activation in the Chl-D2RKO is restored by blocking the M4R by tropicamide, a specific antagonist (Lewis et al. 2020). As tangible evidence of M4R activation in the Chl-D2RKO, in this article, we show by chromatin immunoprecipitation that M4R antagonism by tropicamide restores the enrichment of H4K8ac at the promoters of IEGs and M4R in the Chl-D2RKO mice to the level of WT. We hope that these data make the case for the M4R overactivation in D1R⁺MSNs of the mutants.

The authors compared WT/Chl-D2R KO groups after cocaine IVSA and then performed RNA-seq, and they also stated Kat5/M4R would be crucial in the development of motivated responses for cocaine, but not positive reinforcement. However, in the following data, they mostly performed acute-cocaine administration and compared the data to observe the role of Kat5/M4R in cocaine-mediated responses. They should carry out IVSA to explore the attributes of M4R/Kat5 to cocaine motivation clearly. These should be resolved.

- We have well taken this point. After the initial identification of the genes by IVSA, we tested whether the effects observed in these conditions appeared only after the acquisition of SA or if this mechanism could be observed also in acute and chronic treatments. This was done to unravel potential mechanisms appearing only after SA. However, KAT5, H4K8ac and M4R induction were also observed after both contingent acute and chronic cocaine regimens, indicating that these changes are not restricted to IVSA. This observation strongly facilitated our studies since IVSA was made in the lab of our collaborator Dr. Fowler, restricting the possibility to use their equipment to repeat these experiments for an extensive amount of time.

Target mechanisms were assayed and validated mainly by perfusion of various drugs (NU9056, tropicamide, and VU04567154), all of which were administered via intraperitoneal (IP) injection. However, it would be questionable whether the same effects could be observed if these drugs were administered in a brain-region-specific or cell-type-specific manner. Additionally, except VU04567154, the effects of those drugs were only tested through locomotion and cocaine-conditioned place preference (CPP). Since cocaine self-administration is a most direct method for assessing cocaine motivation as they reckoned, they should examine whether these drugs have any impact on motivation using IVSA.

- We have well taken this point, and the answer is similar to the previous one. Regarding the region-specific effects of the compounds used in this study, intracerebral injections in IVSA cannulated mice are not impossible, while it would be challenging to perform this complex of an experiment using collaborators' equipment. Here again, in response to this referee's criticism, we are confident that the results obtained with the various drugs in both acute and chronic cocaine experiments could be extended to the IVSA. This, not because the effects of contingent administration are similar to that of non-contingent, but because at the molecular level we found that Kat5, H4K8ac and M4R are similarly regulated by the different regimens. Furthermore, in order to exclude any non-specific effects (unrelated to absence of D2R in ChIs) of the administered drugs, we always compare the phenotype of mutant mice to that of WT littermates used in parallel in the presence or absence of the different compounds. The results obtained between genotypes from our behavioral studies are highly significant, which we trust are indicative of the drugs' specific effects.

There are a number of minor points as follows:

Fig 1. The authors have not confirmed whether the knockout effect of Ch1-DRD2KO mouse was confined to the NAc. Experiments for region-specific deletion of DRD2 in Ch1, or rescuing overexpression of DRD2 in the NAc of Ch1- D2RKO mice would be required.

- This reviewer is correct in that we have used Ch1-D2RKO mice, thus the knockout of D2R is not restricted to the NAcc but to all striatal cholinergic interneurons (Kharkwal et al. 2016). However, the cholinergic interneurons are known to be interconnected and therefore even injecting in the accumbens it would not be possible to exclude a global striatal effect. For instance, in previous experiments using a DREADD virus injected in the ventral striatum to silence ChIs, we obtained same results than that of Ch1-D2RKO mice receiving a systemic injection of tropicamide (Lewis et al. 2020). Furthermore, targeting the cholinergic interneurons using viral vectors is, to my knowledge, not possible since the ChAT promoter cannot be introduced into viral vectors.

Fig 2. When observing transcriptomic changes after PR, the number of cocaine

infusions was already different. This would necessitate additional experiment with control group that received the same amount of cocaine using a non-contingent paradigm.

- We appreciate the reviewer's comment regarding the differences in the number of cocaine infusions between WT and KO mice on the day of the progressive ratio (PR) session and the suggestion to include a control group using a non-contingent paradigm.

Our study's design aimed to evaluate transcriptomic changes associated with motivation for cocaine by selecting mice that had reached a comparable level of self-administration behavior prior to the PR session, rather than equating total cocaine exposure across groups. This approach allowed us to capture gene expression differences that may be related to differences in motivation rather than total cocaine intake.

Thus, RNA expression analysis and comparison between WT and Chl-D2RKO after PR were the object of this study. We believe the differences observed at this level reflect gene expression in WT mice as compared to mice that are missing the normal dopaminergic inhibitory tone on Chls in the same conditions. Furthermore, we also show that when administered an equal amount of cocaine in both acute and chronic, *Kat5* induction and the series of events downstream of it differ similarly between WT and Chl-D2RKO mice to what we found after IVSA, begging to exclude the total drug intake as reason for these observed differences.

*Fig 4. Since *Kat5* would be expected to work across whole brain, it should be depleted (shown in ref 30, See Allen brain ISH data #69735174) or at least drugs should locally delivered via intracerebral infusion to target brain areas.*

- We have well taken this point, and indeed future experiments will be aimed at selective knockdown *Kat5* in MSNs. While possible, intracerebral infusions of drugs are challenging in the IVSA setting; and what the reviewer proposes would be a whole new study, which we agree could be done in the future. This new study would need a larger number of mice (to consider higher mortality rates and clogging of the probes) and the use of IVSA equipment for longer period of times in the collaborating laboratory.

*Can the effect of the *Kat5* inhibitor be conclusively determined from capturing *Kat5* in western blot? Capturing histone acetylation (substrate of *Kat5*) via western blot or IHC would be more straightforward (shown in ref 30).*

- We have well taken this point and have added multiple new experiments showing the localization and expression of *KAT5*, D1R, D2R and H4K8ac in response to chronic treatment with cocaine in the presence or absence of NU9056 by FISH and FISH coupled to IF (Figs. 7-8).

Reviewer #3 (Remarks to the Author):

The study by Lewis et al. identified deficits in cocaine seeking and cocaine-induced long-lasting molecular changes in the NAc in mice lacking D2R in ChIs. They unveiled that the absence of D2R in ChIs prevents the upregulation of histone acetyltransferase, KAT5 in the NAc of cocaine-treated mice. This event is accompanied by a reduction of the acetylation of histone H4 on lysine 8 normally detected in the promoters of various IEGs. They finally designed a serie of experiments aiming to demonstrate that these effects are due to a hypercholinergic state in the ChI-D2RKO. Overall, the study is well-designed and adds to an important topic in the field. Despite my excitement about the novel insights explored in this study, several issues interfere with the correct interpretation of the findings. Below is a list of additional experiments to be performed by the authors to strengthen the conclusion of the present work.

Major concerns

Intro: Second paragraph : the notion of direct and indirect pathway refers to the dorsal striatum not to the NAc. Accumbal D1R-expressing MSNs do not project to the SN but to the VP and to lesser extent to the VTA. Please correct and quote appropriate references.

- We thank the reviewer for this comment; we have changed the nomenclature in D1R⁺MSNS and D2R⁺MSNs and added the correct references.

Fig 1: It is not clear whether male and/or female mice are used. SABV is neither considered nor is the combination of male and female mice into a single data set justified if applicable. This aspect is crucial considering that behavioral and molecular alterations mediated by cocaine strongly differ between male and female mice (PMID: 34272455 ; PMID: 28072417). This issue should be clearly addressed. In Fig S2B, both wt and ChI-D2RKO press more on the active lever delivering saline. Does it mean that saline self-administration is reinforcing in mice? Please clarify this issue.

- We appreciate the reviewer's comment regarding the consideration of sex as a biological variable. In this study, we used only male mice, consistent with our previous work (Lewis et al. 2020), to ensure continuity in our experimental design and facilitate direct comparisons between studies. To address this limitation, we have revised the manuscript to explicitly state that only male mice were used. However, we acknowledge that behavioral and molecular responses to cocaine can differ between male and female mice, as noted in the mentioned studies. While including both sexes would provide additional insights into sex-specific mechanisms, our goal in this study was to maintain consistency with our established model. The chain of events described in this study would be surprising if only present in males and not in females, but it cannot be excluded.

Future studies will be necessary to determine whether the observed transcriptomic changes generalize to female mice or if sex-specific differences exist. Regarding the second comment about the interpretation of saline self-administration and the observation that both WT and Chl-D2RKO mice exhibited active lever pressing in the absence of cocaine. Prior studies have shown that mice self-administer similar levels of saline when tested with a dose response function, indicating that saline itself is not reinforcing (Fowler & Kenny, 2011).

To further clarify this issue, we performed a statistical analysis of active vs inactive lever presses, this comparison demonstrates that the relative preference for the active lever does not significantly change across sessions, suggesting that saline self-administration does not produce reinforcement but rather reflects residual operant behavior.

Fig 2: Panel B. It is not clear from the text and the volcano plot what these DEGs represent: a difference between saline vs cocaine independently of genotypes or among all genes downregulated by cocaine, 897 genes are different between wt and Chl-D2RKO. Please clarify the text and include additional RNAseq analyses. Related to my previous comments. If both male and female mice are used, how the authors manage to integrate potential sex-differences in their analyses? The analysis in panel D is original but cell-type specific increased expression of Kat5 must be confirmed by smFISH with probes allowing the identification of MSNs, ChIs and other classes of GABAergic interneurons. This should be compared to Chl-D2RKO. This control is crucial if one wants to understand at which circuit level the lack of D2R in Chl impact the expression of Kat5.

- We appreciate the reviewer's comment regarding the interpretation of differentially expressed genes (DEGs) and the need for additional clarification of the analyses. Our primary focus was on identifying genes that were differentially expressed in WT mice in response to cocaine but did not show similar changes in the Chl-D2RKO group in males (Fig. 2b-c). This approach allowed us to investigate genes that are typically responsive to cocaine exposure but may be dysregulated in the absence of Chl-D2R signaling. To further address this concern, we have now included an additional analysis comparing genes that differ in expression between genotypes in the cocaine groups independent of the saline groups (Fig. S4), as per reviewer's 2 request. This new analysis helps delineate transcriptomic differences that are specific to cocaine exposure. We have updated the text in the Results section to clarify these distinctions and modified the volcano plot figure legend to specify what the DEGs represent.

Regarding the consideration of sex as a biological variable (SABV), as mentioned in our previous response, this study exclusively used male mice to maintain consistency with our prior work. This design choice ensures that our findings build directly on our previous results without introducing potential variability due to sex differences.

We have updated the text accordingly to improve clarity regarding our DEG analysis approach and our rationale for using male mice.

Fig 3: The increase of KAT5 expression (at protein level) detected only one hour after a single cocaine administration is impressive. Immunofluorescence analysis of KAT5 expression in D1tomato/EGFP and D2eGFP mice should be performed to complete the characterization. This level of analysis would also precise whether increased KAT5 expression occurs in NAc Core and/or Shell. If the antibody does not allowed such level of analysis, this could be solved by smFISH (see comment Fig 2). Please indicate in the text whether ChIP experiments were performed on NAc extracts from mice acutely administered with cocaine or after IVSA.

- We thank the reviewer for this comment that helped observing the expression of KAT5 mostly in MSNs. Unfortunately, commercially available Kat5 antibodies were unable, in our hands, to give specific signals, despite the use of different antibodies and IF conditions on either cryostat or vibratome sections of brain tissues. We thus performed FISH (RNAscope) analyses using D1R and D2R probes together with a KAT5 specific probe. As you'll see from the quantifications of our results, KAT5 is induced in both MSNs, although significantly only in D1R⁺MSNs. The induction of KAT5 originally observed by RNA-seq analyses of mice undergoing IVSA was also verified and obtained after acute and chronic administration of cocaine. No differences were observed between the core and shell of the NAcc, while we found that Kat5 is similarly regulated in the DLS.

Fig 4: Panel B. It is not clear from the text what these WB represent. If color codes correspond to the ones on panel A then these results deserved to be explained. If I understand well, the increased expression of KAT5 by cocaine (acute or chronic?) depend on the activity of KAT5? If so, the authors should at least demonstrate a H4K8Ac enrichment at the promoter of Kat5 following cocaine administration. In the same line, the authors should demonstrate that the H4K8Ac enrichment at the promoter of Fos, Nr4a1 and Egr1 is causally linked to the increased of KAT5 expression using NU9056.

- We apologize for not being clear on the mechanism leading to KAT5 induction. KAT5 is significantly induced by cocaine in WT D1R⁺MSNs, but not in Chl-D2RKO MSNs. We speculate that Kat5 in D1R⁺MSNs is dependent on the cAMP/PKA pathway. Indeed, Kat5's induction can be restored in the Chl-D2RKO by blocking M4R signaling in D1R⁺MSNs. Thus, indicating that Kat5 induction in D1R⁺MSNs depends on the D2R-mediated inhibition of ChIs and ACh signaling and simultaneously through DA through D1R. Once Kat5 is induced, it acetylates histone H4 on Lysine8 (H4K8ac), which is enriched at IEGs and M4R promoters as shown by ChIP (Fig. 6b). In the revised version, we have performed FISH on animals treated with NU9056 and shown that it abolishes Kat5 induction in the striatum.

Fig 5: *Please confirm the increased of Chrm4 by smFISH to determine in which striatal-cell type this regulation occurs. This result will be important for the interpretation of the data obtained with the M4R antagonist.*

- We thank the reviewer for this suggestion that brought us to perform FISH analysis of M4R induction. Interestingly we found that M4R is induced specifically in the NAcc in D1R⁺MSNs (Fig.5 and S6).

Please precise whether these data have been generated on NAc extracts from mice acutely administered with cocaine. I am not sure that based on these results we can conclude we have a hypercholinergic phenotype.

- The hypercholinergic phenotype is based on pharmacological evidence showing that treatment with the M4R antagonist tropicamide restores the motor and rewarding responses in Ch1-D2RKO mice to WT levels (Lewis et al 2020). In this article, we performed ChIP using H4K8ac antibodies on NAcc extracts from WT and Ch1-D2RKO mice treated in acute with cocaine in the presence or absence of tropicamide (Fig. 6b). Interestingly, we found that the M4R promoter is regulated through the Kat5-dependent acetylation of H4K8ac. With respect to the hypercholinergic phenotype, our hypothesis is based on the pharmacological effects of blocking M4R or Ch1s activity by DREADD as previously shown (Lewis et al. 2020). We have added a sentence in the discussion to underline that a direct measurement of ACh is however needed in support of this hypothesis.

In their reasoning, the authors seem to have totally neglected two important points. The first one refer to the heterogeneity of Ch1s responses due to regional variation in dopamine-neuron glutamate cotransmission “Dopamine neurons drive a burst-pause firing sequence in cholinergic interneurons in the medial shell of the nucleus accumbens, mixed actions in the accumbens core, and a pause in the dorsal striatum.” (PMID: 24559678). The second important aspect is related to the fact that Ch1s express VGLUT3 and use both acetylcholine and glutamate as neurotransmitters, a property crucial for behavioral and cellular effects mediated by cocaine (PMID: 26239290). Knowing the importance of this co-transmission, the impact of the absence of D2R signaling in Ch1s on the Ach/Glu co-transmission should be addressed and integrated in the interpretation of results.

- We have well taken this comment. The striatal circuits are complex and while by electrophysiological means it is possible to study the effects of specific signaling at the single neuron level, this becomes arduous when analyzing behavior. We totally agree with the reviewer on the complexity of dopamine neuron-mediated control of the striatum, which includes the release of glutamate together with dopamine. There are also other sources of glutamate, and we think in particular to the corticostriatal fibers which also express M4R. We have added a short paragraph in the discussion, according to this criticism. Nevertheless, in this article, our focus has been on the ACh/DA interactions, since our knockout was cell-specific and directed to the Ch1s. We also recognize that Ch1s have been

reported to release glutamate, however the pharmacological studies performed in vivo appeared to suggest an ACh specific effect on the behavioral phenotype that we observed. We have done our best to address this point in the discussion.

Fig 6 & 7: CPP measures the rewarding property of a drug not the reinforcing property (p12 line 276 / p13 line283 please correct). Reinforcement refer to instrumental learning. IVSA allows the evaluation of the reinforcing property of cocaine. In Fig 7 panel C, the breaking point in control mice is much lower than the ne reported in Fig 1. Could the authors comment on this? The authors should be more cautious with their conclusion. They cannot infer what happens in Ch1-D2RKO based on what they observed in control mice treated with M4R PAM and without addressing the co-transmission issue (see above).

- We appreciate the reviewer's careful assessment of our terminology and have corrected references to the rewarding and reinforcing properties of cocaine to ensure accuracy. These corrections have been implemented in the relevant sections of the manuscript. Regarding the difference in breaking points between control mice in Fig. 7C and those reported in Fig. 1, this discrepancy is due to cohort effects and/or differences in training duration between the two experiments. In the first experiment (Fig. 1), mice underwent self-administration training over three weeks, whereas in the second experiment (Fig. 7C), training was conducted over two weeks. Variability in breaking points across different cohorts is well-documented in self-administration studies and can result from differences in training duration, handling, and individual variability in drug-seeking behavior. We have now included a discussion of this potential cohort effect in the revised manuscript. Additionally, we acknowledge the reviewer's concern regarding our interpretation of the Ch1-D2RKO results and the potential role of co-transmission. While our findings in control mice treated with the M4R PAM suggest a possible mechanism, we recognize that additional studies are needed to directly address co-transmission effects in Ch1-D2RKO mice. We have revised the manuscript to be more cautious in our conclusions and have explicitly stated this limitation in the discussion.

Discussion should be expanded to address raised inconsistencies in the Results. However, it is advisable to rewrite the discussion regarding the behavioral experiments, pending suggested experiments and analyses.

The Results and Discussion have been revised according to the new results obtained and in response to criticisms. We sincerely hope that the reviewer would be satisfied of these changes.

Minor concerns

- p3 line 60 : Pease quote papers related to mouse nucleus accumbens. The present

refs are not really appropriate.

- p8 line 174: Figure 2A should be called not 2C. Please correct.

- p10 line 216: Figure miscalled. Fig 3B should be called not 4B. Same changes should be done for the paragraph.

- p12 line256: Figure miscalled. Fig 5B should be called not 4B. Same changes should be done for the paragraph.

The figures and their order have been changed. We carefully reviewed the naming of all of them.

In bold the reviewers' notes

Reviewer #1

I was very enthusiastic about the original version of this study and find the revised version that much stronger. I thank the authors for attending to cell-type-specific mechanisms of Kat5. The manuscript is acceptable for publication in its present form and will be a strong contribution to the literature.

We thank this reviewer for suggesting the experiments now present in the revised article and for recognizing the importance of our findings.

Reviewer #2:

Thank you for your thoughtful revisions. The manuscript has improved, but some key concerns regarding mechanistic specificity and causality in the proposed epigenetic-behavior model remain unresolved.

We thank the reviewer for recognizing the efforts made to improve the article following the suggestions sent with the first revision of our manuscript. Despite this reviewer opinion, we believe that the mechanism underlying the effects observed in our Ch1-D2RKO mutants in response to cocaine has been well delineated.

Below are the major points requiring attention:

1. While the RNAscope data (Figs. 5, 7) elegantly demonstrate correlative upregulation of Kat5 and M4R in D1R+ MSNs, the claim of cell-type-specific necessity lacks functional validation.

To strengthen causality, this reviewer suggests:

- 1) Cell-type-specific deletion/depletion of Kat5 (e.g., using D1-Cre or D1-FLP mice).**
- 2) Rescue experiments (e.g., M4R overexpression in D1-MSNs to test sufficiency).**
- 3) GRAB_ACh sensor data to quantify dynamic ACh changes in the striatum.**
- 4) HPLC-based ACh measurements in Ch1N-D2R KO mice under relevant conditions.**

The requests of this reviewer are beyond all reason. The four points could be used for a five-year grant's proposal based on the publication of the data contained in our article! Unfortunately, this is not a fair review, and we beg to differ from this reviewer view of the study.

These experiments would directly test whether Kat5/M4R functions in D1-MSNs would be selective and necessary for the observed phenotypes.

In our article we have used the inhibitor of Kat5 and the M4R antagonist as well as a positive allosteric modulator comparing wild-type and mutants and found support to our findings.

Furthermore, the increase of M4R is observed in MSN-D1R+ of wild-type but not Chl-D2RKO mice, as observed by RNAscope analyses indicating that these neurons play a central role. Now, it cannot be excluded that other genes might be implicated, but this would be the aim of future studies.

We identified H4K8ac as cocaine and Kat5 induced modification in the NAcc, shown by western, RNAscope-IF and ChIP. Importantly, ChIP shows an enrichment of this modification on the promoter of M4R as well as of the immediate early genes in the NAcc. Thus, it is not fair to state that our study lacks specificity and is only correlative.

We want to emphasize that the focus of the article is the consequence of loss of the D2R control on the cholinergic interneurons in the contingent and non-contingent response to cocaine.

2. The model hinges on the premise that D2R deletion in cholinergic interneurons (ChINs) elevates tonic ACh release, but direct evidence is missing. Without this, the link between D2R loss, ACh release, and M4R activation remains speculative.

How it can be speculative if the phenotype of the Chl-D2RKO mice can be reverted by tropicamide, while the wild-type can act as mutants if given a positive allosteric modulator (PAM) of M4R? Why the pharmacological studies present in the article are not considered?

3. The behavioral effects of M4R/Kat5 manipulation are compelling but do not yet prove that these molecular changes are necessary or sufficient in imparting the observed phenotypes to the ChIN-D2R KO mice.

Please refer to the point made above where pharmacological treatments can rescue the mutants while the PAM induces it in the wild-type.

Moreover, the epigenetic and behavioral consequences seem consistent with their observation but remain speculative. Addressing these questions with additional experimental data would solidify the proposed mechanistic chain.

The epigenetic and behavioral consequences do not "seem" they **are** consistent with our results.

What is the limit for additional experimental data? The present version contains 8 figures and 7 supplemental figures.

The reviewer knows that there are no limits to produce data and that each new result calls for new experiments. The implication of dopamine in substance use disorders is known since decades, nevertheless we are still working on understanding the mechanisms underlying it. Here we offer evidence of novel candidates implicated in the control of cocaine effects and intake. As per reviewer 1, these findings will make a strong contribution to the field.